# MERGE³: Efficient Evolutionary Merging on Consumer-grade GPUs

**Tommaso Mencattini** [* 1]  **Adrian Robert Minut** [* 2]  **Donato Crisostomi** [2]  **Andrea Santilli** [2]  **Emanuele Rodolà** [2]

## Abstract

Evolutionary model merging enables the creation of high-performing multi-task models but remains computationally prohibitive for consumer hardware. We introduce MERGE³, an efficient framework that makes evolutionary merging of Large Language Models (LLMs) feasible on a single GPU by reducing fitness computation costs 50× while retaining a large fraction of the original performance. MERGE³ achieves this by **E**xtracting a reduced dataset for evaluation, **E**stimating model abilities using Item Response Theory (IRT), and **E**volving optimal merges via IRT-based performance estimators. Our method enables state-of-the-art multilingual and cross-lingual merging, transferring knowledge across languages with significantly lower computational overhead. We provide theoretical guarantees and an open-source library, democratizing high-quality model merging.

github.com/tommasomncttn/merge3

## 1. Introduction

Model merging has become a powerful and accessible approach for developing new state-of-the-art models without the need for cluster-grade computing typically required for large model training (Yang et al., 2024a). Its key advantage lies in performing the merging process post-hoc directly in the parameters of endpoint models—that is, pre-existing models (either base or fine-tuned) that serve as the components of the merging process—eliminating the need for training and significantly reducing the demand for expensive computational resources.

---
[*]Equal contribution [1]School of Computer and Communication Science, Ecole Polytechnique Fédérale de Lausanne, Lausanne, Switzerland [2]Department of Computer Science, Sapienza University of Rome, Rome, Italy. Correspondence to: Tommaso Mencattini <tommaso.mencattini@epfl.ch>.

*Proceedings of the 42nd International Conference on Machine Learning*, Vancouver, Canada. PMLR 267, 2025. Copyright 2025 by the author(s).

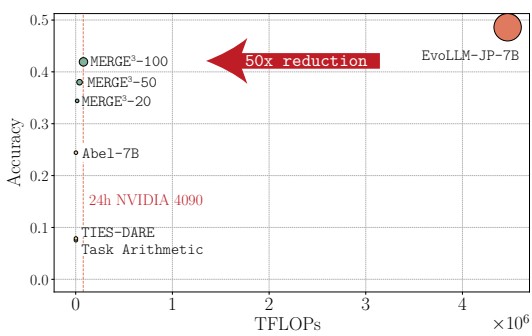

*Figure 1.* **Accuracy on Japanese `GSM8K` over fitness evaluation FLOPs.** MERGE³ is competitive with a model evolved on the full dataset by only using a consumer-grade GPU and 2% of the data (point size reflects data amount).

This approach has significantly broadened access to the field, with ML practitioners producing competitive models out of existing ones on standard consumer GPUs[1](Ilharco et al., 2022). However, although computationally inexpensive, most of the existing approaches are quite rudimentary, require ad-hoc choices, and are usually based on ungrounded trial-and-error strategies for selecting the merge coefficients, which ultimately limits their downstream performance (Yadav et al., 2023; Yu et al., 2024). On the other hand, recent work has shown that evolutionary merging can produce models of unprecedented quality by automating the hyperparameter search for merging coefficients (Akiba et al., 2025). While this technique can incorporate any standard merging method, such models are absent from public leaderboards likely due to a mismatch between the high computational demands of evolutionary merging and single-GPU setups typical of merging practitioners. Indeed this computational cost is significantly high: computing the fitness function requires generating and evaluating answers for each dataset element, for each candidate in every evolutionary step. As shown in Figure 1, the fitness computation alone in the 1,000-trial evolutionary merge from Akiba et al. (2025) requires approximately $4 \times 10^6$ TFLOPs, with the full algorithm demanding largely over a month of

---
[1]At the time of writing, around 30% of models on the Hugging Face Open LLM leaderboard are merged models.

continuous computation if run on a single NVIDIA 4090 (§C.3.3) with 24 GB of VRAM. Requiring repeated and costly runs of large language models, the fitness evaluation is the primary bottleneck. This makes evolutionary merging out of reach on consumer hardware, potentially excluding the very users it was meant to empower.

In this paper, we address this challenge by introducing MERGE[3], an evolutionary merging framework that runs on a single consumer GPU with competitive results (see fig. 1). Unlike the competing approach, MERGE[3] operates with just $0.077 \times 10^6$ TFLOPs, namely a **50-fold reduction**. This drastic decrease in computational cost makes it feasible on consumer hardware, freeing up FLOPs for further optimization or additional tasks.

Our approach starts by **E**xtracting a reduced subset of the fitness evaluation dataset, significantly alleviating the computational bottleneck of fitness computation (fig. 2). However, this reduction risks losing accuracy if the subset lacks diversity. To address this, we apply Item Response Theory (IRT) (Lord et al., 1968)—a well-established statistical framework—to bridge the gap between reduced-dataset evaluations and full-dataset performance. Specifically, we first **E**stimate the latent abilities of the endpoint models using IRT, ensuring the merged models accurately reflect their components' strengths. Then, we **E**volve the endpoint models with IRT-based performance estimators designed for model merging, assuming the merged model's ability is a combination of those of the endpoint models. This approach significantly improves the efficiency and accuracy of fitness estimation, integrating merging-specific insights into performance estimation theory while maintaining high accuracy with reduced datasets.

Experimental results show that MERGE[3] effectively transfers mathematical skills by merging a strong math model with three language-specific models, achieving 10–20% higher accuracy than standard merging baselines in each language. Building on this, we evolve a single multilingual model by merging Italian, English, German, and Dutch models, outperforming individually fine-tuned models by up to 19% on `ARC` (Clark et al., 2018), a widely used benchmark for reasoning. Furthermore, MERGE[3] achieves competitive accuracy on Japanese `GSM8K` (Cobbe et al., 2021), matching models evolved on full datasets while maintaining high efficiency, demonstrating that our evolutionary strategy preserves performance while drastically reducing computational costs.

To summarize, our contributions are fourfold:

- We introduce a novel, efficient evolutionary model merging framework leveraging Item Response Theory, making merging feasible on consumer hardware.

- We demonstrate its effectiveness in transferring skills

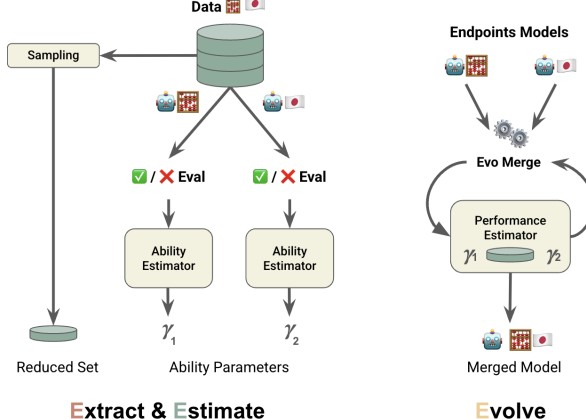

*Figure 2.* **MERGE[3] for math + Japanese merging (`GSM8K`)**. The method **Extracts** a reduced evolutionary dataset, **Estimates** ability parameters ($\gamma$) via Item Response Theory (IRT) based on their response correctness, and **Evolves** the endpoint models through iterative merging. Leveraging an IRT-based performance estimator, it approximates full-dataset fitness with reduced data, cutting fitness estimation costs while preserving full-dataset accuracy – making evolutionary merging feasible on consumer GPUs.

across languages and synthesizing state-of-the-art multilingual models without standard training.

- We advance the theoretical foundations of performance estimation in model merging and provide formal guarantees for our proposed estimators.

- We release a modular library for evolutionary merging on consumer GPUs, alongside a suite of state-of-the-art models for several low-resource languages.

## 2. Related Work

**Model Merging** has emerged as an efficient alternative to ensembling by integrating existing models without any additional training. One set of methods identifies neuron permutations that align the models into a shared optimization basin, allowing them to be merged through straightforward averaging (Ainsworth et al., 2022; Jordan et al., 2023; Stoica et al.; Peña et al., 2023; Crisostomi et al., 2025). Closer to our work, multi-task model merging focuses on the case where a single pre-trained model is fine-tuned for different tasks (Ilharco et al., 2022; Yadav et al., 2023; Yu et al., 2024; Matena & Raffel; Wortsman et al., 2022; Davari & Belilovsky, 2025; Wang et al., 2024; Zhou et al., 2024; Gargiulo et al., 2025). In this direction, several works address task interference by pruning or selectively combining parameters—e.g., TIES-merging (Yadav et al., 2023), Model Breadcrumbs (Davari & Belilovsky, 2025), and DARE Merging (Yu et al., 2024)—or by opti-

mizing merge coefficients (Yang et al.), introducing task-specific modules (Yang et al., 2024b), and disentangling weights (Ortiz-Jimenez et al., 2024).

**Evolutionary Algorithms.** Evolutionary Algorithms are black-box optimization algorithms operating on a population of potential solutions by evolving them through generations with operators such as selection, mutation, recombination, and crossover (Bäck & Schwefel, 1993; Pétrowski & Ben-Hamida, 2017; Dasgupta & Michalewicz, 1997). Recent applications include neural architecture search (Real et al., 2019) and hyperparameter tuning (Vincent & Jidesh, 2023), where evolutionary methods efficiently navigate large design spaces without manual intervention. The fitness function is crucial, as it evaluates the quality of each solution, guiding the selection process by favoring higher-scoring (fitter) solutions for reproduction (Eiben & Smith, 2015). Closest to our work, Akiba et al. (2025) propose to apply evolutionary algorithms to optimize model merging recipes, eliminating the need for trial-and-error in combining parameters. In this context, the most obvious candidate for a fitness function is simply the performance of the resulting model over a held-out validation set.

**Item Response Theory.** Item Response Theory (IRT) (Cai et al., 2016; Van der Linden, 2018; Brzezińska, 2020; Lord et al., 1968) is a paradigm to design, analyze, and score responses to tests such as SAT or GRE (An & Yung, 2014; Kingston & Dorans, 1982; Petersen et al., 1982). Based on the relationship between individuals' performances on a test item and the test takers' levels of performance on the corresponding required ability, IRT has recently spread from psychometrics to natural language processing. In this direction, Lalor et al. (2016) leverage IRT's latent dimensions to evaluate language models, while Vania et al. (2021) use it to analyze benchmark saturation in NLP evaluations. More relevant to our work, Zhuang et al. (2023) and Polo et al. (2024) employ IRT-driven adaptive testing to alleviate the computational burden of large-scale evaluations for large language models (LLMs). Although their focus is on LLM evaluation, which shares similarities with the efficient evaluation of fitness functions in model merging, our work builds on these approaches to design IRT-based estimators specifically tailored for model merging. Unlike prior applications of IRT, which are limited to LLM evaluations, our approach adapts the framework to address the unique challenges of evolutionary model merging, enabling efficient and accurate fitness estimation.

# 3. MERGE³

Our method MERGE³ speeds up evolutionary model merging by reducing the computational cost of fitness eval-

uation. It achieves this by shrinking the fitness evaluation dataset and using IRT-based performance estimators to maintain full-dataset accuracy from subset evaluations. Figure 2 shows an overview of our method, while we present below the pseudo-code for the end-to-end MERGE³ algorithm.

---

**Algorithm 1** The full MERGE³ algorithm.

---

**Require:** Dataset $D$, models $\{M_1, M_2, \ldots, M_n\}$, iterations $T$
**Ensure:** Pareto-optimal merged models
1: $\bar{D} \leftarrow \text{RANDOMSAMPLE}(D, k)$     # Sample $k$ items from $D$
2: $\{\gamma_1, \ldots, \gamma_n\} \leftarrow \text{ESTIMATEABILITIES}(\{M_1, \ldots, M_n\}, D)$
3: $P \leftarrow \text{GenerateInitialPopulation}\{M_1, \ldots, M_n\}$
4: **for** $t \leftarrow 1$ to $T$ **do**
5:     **for all** $M \in P$ **do**
6:        $\lambda \leftarrow \text{FITLAMBDA}(M, \{\gamma_1, \ldots, \gamma_n\}, \bar{D})$
7:        preds $\leftarrow \text{GETPREDICTIONS}(M, \bar{D}, \lambda)$
8:        corr $\leftarrow \text{GETCORRECTNESS}(\text{preds}, \bar{D})$
9:        $F(M) \leftarrow \text{ESTIMATEFITNESS}(\text{corr}, \lambda)$
10:    **end for**
11:    $P \leftarrow \text{SELECTPARENTS}(P, f)$     # Select based on fitness
12:    $P \leftarrow \text{APPLYMUTATION}(P)$
13:    $P \leftarrow \text{APPLYCROSSOVER}(P)$     # Generate offspring
14: **end for**
15: **return** $\text{PARETOFRONT}(P)$

---

## 3.1. Extract & Estimate

Evaluating the fitness function involves generating and assessing answers for each data sample, repeated across all models in the population at every evolutionary step. Given the computational demands of evolutionary algorithms and LLMs, this process is highly intensive. To mitigate this, we reduce the dataset $D$ to a smaller subset $\bar{D} \subset D$ with $|\bar{D}| \ll |D|$. After exploring various subsampling strategies, we found uniform random sampling as effective as more complex methods (see appendix C.1) and adopted it for simplicity. Since dataset reduction is not our main focus, we leave further optimizations for future work.

Reducing the dataset speeds up evaluation but does not guarantee identical results – particularly when the subset is significantly smaller, as in our case. To bridge this gap, we build an IRT-based estimator that adjusts for this discrepancy, effectively estimating performance to reflect full-dataset results (Lord et al., 1968; Polo et al., 2024).

**IRT model.** We first define an estimator to assess each endpoint model's inherent abilities, derived from the latents of a Bayesian network. This ensures that merging preserves individual model strengths. In the Evolve step (§3.2), the estimated latent abilities are fed to a *performance* estimator to compute the final fitness.

To estimate LLM abilities, we build on Polo et al. (2024), who applied IRT to evaluate LLM performance; however,

while they used IRT for benchmarking, we extend it to estimate inherent abilities relevant for model merging, and explicitly use them to guide merging in the Evolve step.

In IRT, latent variables ($\gamma$) represent a model's underlying abilities, while manifest variables ($Y$) indicate response correctness. The framework models the probability of a correct response based on model abilities and item characteristics (e.g., difficulty).

IRT defines this probability as:

$$\mathbb{P}(Y_{im} = 1 \mid \gamma_m, \alpha_i, \beta_i) = \frac{1}{1 + \exp(-\alpha_i^\top \gamma_m + \beta_i)} \quad (1)$$

Here, $\gamma_m \in \mathbb{R}^d$ represents model $m$'s latent abilities, $\alpha_i \in \mathbb{R}^d$ defines the ability dimensions needed to answer example $i$, and $\beta_i$ denotes its difficulty. A model is more likely to answer correctly when its abilities ($\gamma_m$) align with the example's required traits ($\alpha_i$) and less likely when the difficulty ($\beta_i$) is higher. $Y_{im}$ is a binary variable indicating whether model $m$ correctly predicts example $i$ (1 if correct, 0 otherwise).

Crucially, this approach estimates a model's likelihood of answering correctly *without directly analyzing the example's content*, relying solely on the estimated IRT parameters ($\gamma_m, \alpha_i, \beta_i$).

**Fitting.** We use variational inference to efficiently estimate both example-specific ($\alpha_i, \beta_i$) and model-specific ($\gamma_m$) parameters within a hierarchical Bayesian model (Lalor & Rodriguez, 2023), initialized as detailed in appendix B.1. Following Polo et al. (2024), we estimate $\alpha_i$ and $\beta_i$ using correctness data ($Y_{im}$) from publicly available model evaluations, namely the Open LLM leaderboard. To estimate $\gamma_m$, each endpoint model generates answers for the full evaluation dataset, which are then used to assess correctness ($Y_i$) (see Figure 2). This procedure is repeated for each model $m$, producing the corresponding $\gamma_m$ ($\gamma_1$ and $\gamma_2$ in the Figure).

To summarize, unlike previous work, where IRT latent abilities remain hidden variables, we explicitly derive $\gamma_m$ as an *ability estimator* to quantify each model's strengths. Additionally, rather than estimating $\gamma_m$ from a subset, we compute it using the *full* evaluation dataset, providing a more comprehensive measure of model ability, which we now leverage to enhance the merging process.

### 3.2. Evolve: Performance Estimator

The *performance estimator*, a key part of the Evolve step, efficiently approximates the fitness function, which measures the merged model's accuracy. Since fitness evaluation runs repeatedly during evolution (once per model per iteration), reducing its computational cost is crucial. In-

stead of evaluating the full dataset, the estimator predicts performance using only the endpoint models' abilities and the reduced dataset from previous steps, significantly accelerating the process.

We introduce two novel performance estimators for merging: merged performance Item Response Theory estimator (MP-IRT) and generalized merged performance Item Response Theory estimator (GMP-IRT). Since model merging linearly combines weights, we assume that the latent abilities of the merged model (e.g., problem-solving or linguistic capabilities) are also a linear combination of the endpoints' abilities. This makes our approach far more efficient, estimating only the interpolation coefficients ($\lambda_i$) instead of recomputing the full ability vector $\gamma$ of the merged model from scratch (as done in P-IRT and GP-IRT (Polo et al., 2024)).

**Assumption 1** (Linear Combination of Latent Abilities). *Let $\{m_0, m_1, \ldots, m_n\}$ be endpoint models with latent ability vectors $\gamma_i$. If a new model $\tilde{m}$ is formed as a linear combination of their parameters, its ability vector $\gamma_{\tilde{m}}$ can be expressed as:*

$$\gamma_{\tilde{m}} = \sum_{j=1}^{n} \lambda_j \gamma_j = [\gamma_1, \ldots, \gamma_n] \lambda \quad (2)$$

*where $\lambda = (\lambda_1, \ldots, \lambda_n)$ are the interpolation coefficients.*

This assumption allows us to compute the multidimensional IRT model (Eq. 1) for model merging as a linear combination of the individual models' abilities:

$$p_{i\tilde{m}} = \mathbb{P}(Y_{i\tilde{m}} = 1 \mid \lambda_1 \gamma_1 + \lambda_2 \gamma_2, \, \alpha_i, \, \beta_i)$$
$$= \frac{1}{1 + \exp\left(-\alpha_i^\top \left(\lambda_1 \gamma_1 + \lambda_2 \gamma_2\right) + \beta_i\right)} \quad (3)$$

Since the endpoint models' latent abilities $\gamma_j$ were preestimated over the full dataset $D$ in the Estimate step, we only need the subset $\bar{D}$ to estimate the interpolation coefficients $\lambda_j$ via MLE.

**Performance Estimators.** To estimate the accuracy of the merged model $\tilde{m}$ using only the reduced dataset $\bar{D}$ and $p_{i\tilde{m}}$, we define the *merged performance-IRT* (MP-IRT) estimator as:

$$\hat{Z}_{\tilde{m}}^{\text{mp-IRT}} = \frac{\hat{\tau}}{|\bar{D}|} \sum_{i \in \bar{D}} Y_{i\tilde{m}} + \frac{1 - \hat{\tau}}{|D \setminus \bar{D}|} \sum_{i \in D \setminus \bar{D}} \hat{p}_{i\tilde{m}} \quad (4)$$

where $\hat{\tau} = \frac{|\bar{D}|}{|D|}$ downweights smaller subsets that may be noisier. In practice, we are considering the observed correctness for the data points we have access to, while $\hat{p}_{i\tilde{m}}$ predictions are used for the rest, enabling accurate performance estimation across all examples despite evaluating only a subset, where $\hat{p}_{i\tilde{m}} =$

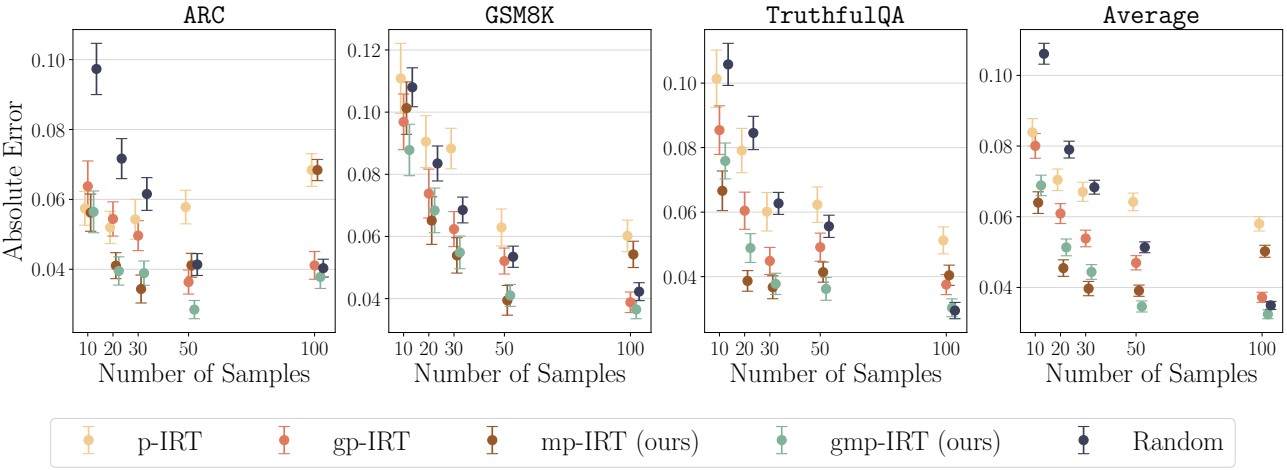

Figure 3. *Performance Estimators:* Absolute error of various estimators as a function of sample size (lower is better). Our MP-IRT and GMP-IRT estimators consistently achieve lower error across various sample sizes and datasets. Additional results available in Figure 13.

$\mathbb{P}\Big(Y_{i\tilde{m}} = 1 \Big| \hat{\lambda}_1\hat{\gamma}_1 + \hat{\lambda}_2\hat{\gamma}_2, \, \hat{\alpha}_i, \, \hat{\beta}_i\Big)$ is the distribution defined by plugging into eq. (3) the parameter found via MLE.

Although designed for model merging, $\hat{Z}_{\tilde{m}}^{\text{mp-IRT}}$ inherits certain limitations of P-IRT (Polo et al., 2024), such as non-uniform weighting and imperfect IRT fits. To mitigate these, we define a *generalized* estimator that interpolates between $\hat{Z}_{\tilde{m}}^{\text{mp-IRT}}$ and the observed correctness on $\bar{D}$:

$$\hat{Z}_{\tilde{m}}^{\text{gmp-IRT}} = c \sum_{i \in \bar{D}} w_i \hat{Y}_{i\tilde{m}} + (1 - c)\, \hat{Z}_{\tilde{m}}^{\text{mp-IRT}} \quad (5)$$

where $c$ is a heuristic scalar and $w_i$ are uniform per-sample weights. We discuss in appendix C.2.3 the optimal choice for $c$. Although model merging can sometimes degrade performance due to weight interference—suggesting non-linear ability interactions— our assumption is empirically supported as we are interested only in evolved models that show a positive performance gain. As validated in our experiments (§4.1), our custom estimators, designed around this assumption, outperform standard IRT estimators.

### 3.3. Evolve: Evolutionary Search

The final step of our algorithm frames model merging as a multi-objective optimization problem. Each merging objective $F(\tilde{m}, D_i)$ represents the performance of the merged model $\tilde{m}$ on task $i$. In practice, we select a multi-objective evolutionary algorithm (e.g., NSGA-II (Deb et al., 2002)) and a merging strategy (e.g., TIES (Yadav et al., 2023)), aiming to optimize the corresponding Pareto front, formally defined as:

$$P_{\overline{F}_D}(\Theta) = \big\{\theta_i \in \Theta : \nexists\, \theta_j \in \Theta \text{ s.t. } \theta_j \succ \theta_i\big\}$$

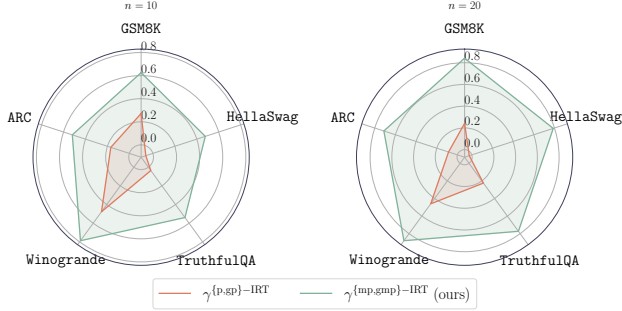

Figure 4. *Ability Estimator:* Cosine similarity between estimated and true abilities for different tasks (higher is better). Our estimated abilities $\gamma^{\{\text{mp,gmp}\}-\text{IRT}}$ better approximate true abilities.

where $\succ$ denotes *Pareto-dominance*. A model $m$ Pareto-dominates $m'$ if:

$$\forall F \in \overline{F}_D : F(m; D) \le F(m'; D)$$
$$\text{and}$$
$$\exists F \in \overline{F}_D : F(m; D) < F(m'; D)$$

This means $m$ is strictly better in at least one metric and no worse in all others. Models on the Pareto front are thus not dominated by any other model.

In our setting, to reduce computational costs, we approximate optimization using $\overline{F}_{\bar{D}}$ instead of $\overline{F}_D$, where $\bar{D} \subset D$ is obtained by the *extraction* step. Performance on $\bar{D}$ is then estimated using the performance estimator.

## 4. Experiments

In this section, we evaluate MERGE[3], demonstrating its effectiveness in evolutionary model merging on consumer-

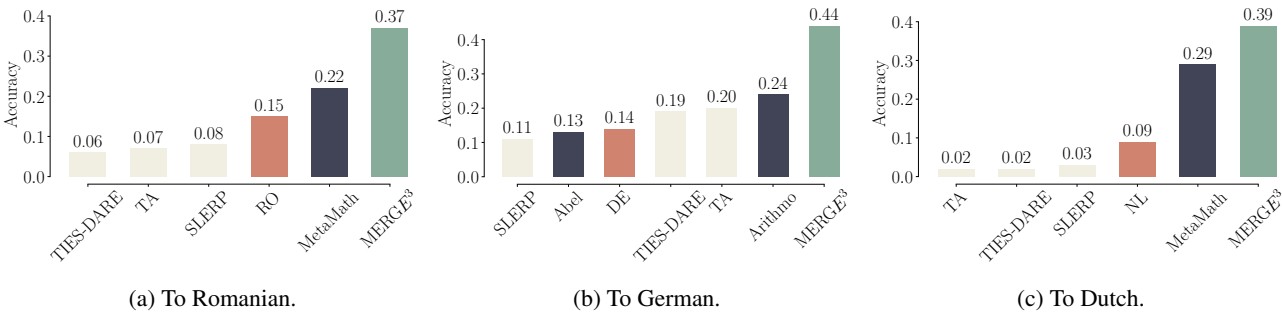

(a) To Romanian.     (b) To German.     (c) To Dutch.

*Figure 5. Cross-lingual skill transfer*: merging math models (dark blue) with language-specific models (red) effectively transfers mathematical skills across languages (green - our method) compared to baselines (white). Accuracy on `GSM8K` for each target language.

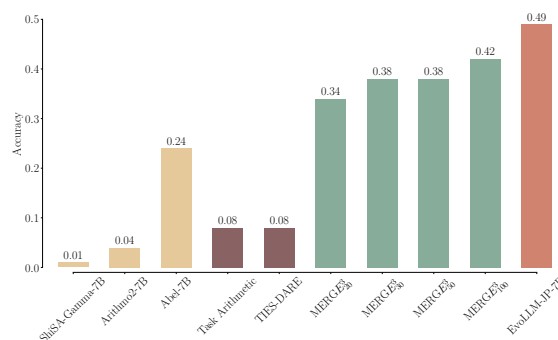

*Figure 6.* Accuracy of merged models for Japanese `GSM8K`.

grade GPUs. We first validate the proposed ability and performance estimators, assessing their accuracy in approximating full-dataset evaluations. Next, we examine cross-lingual transfer, where MERGE³ enables efficient merging of multilingual models, improving mathematical reasoning across languages. Thereafter, we evaluate its ability to synthesize multilingual models, surpassing individual fine-tuned baselines while remaining computationally efficient. Finally, we analyze the performance of MERGE³ on different GPUs. All the merging experiments were performed with our custom-made library *Mergenetic* (see Appendix A) on a RTX 4090 GPU featuring 24 GB of VRAM, while employing a batch size of 8, 4-bit quantization, and models comprising ≈ 7 billion parameters (see Appendix B).

### 4.1. Validating Estimators

In this section, we empirically validate our merged-performance estimators by comparing them against standard P-IRT and GP-IRT estimators (Polo et al., 2024) across five benchmark datasets: `GSM8K` (Cobbe et al., 2021), `Winogrande` (Sakaguchi et al., 2021), `TruthfulQA` (Lin et al., 2022), `Hellaswag` (Zellers et al., 2019), and `ARC` (Clark et al., 2018). Due to space limitations, additional results are provided in Appendix C.

**Ability Estimators.** To validate our ability estimators we compare their inferred latent ability vectors to the reference "ground-truth" vectors $\Gamma$. Specifically, we measure the cosine similarity and the Euclidean distance from the ground-truth $\Gamma$ both for $\gamma^{\{\mathrm{mp,gmp}\}-\mathrm{IRT}}$, estimated with our merged-performance IRT approaches, and $\gamma^{\{\mathrm{p,gp}\}-\mathrm{IRT}}$, estimated with the P-IRT and GP-IRT estimators (Polo et al., 2024). Here, $\Gamma_m$ is computed by fitting the IRT model (as in section 3.1) to each merged model $m$ using its entire set of responses on the full dataset $D$. Incorporating all available data, $\Gamma_m$ serves as our best proxy for the model's true ability. Conversely, both $\gamma_m^{\{\mathrm{mp,gmp}\}-\mathrm{IRT}}$ and $\gamma_m^{\{\mathrm{p,gp}\}-\mathrm{IRT}}$ are estimated using only a smaller subset $\bar{D} \subset D$ of size $n$. Figure 4 shows the results of this comparison for $n = 10$ and $n = 20$, while the results for $n = 15, 30, 50, 100$ are reported in Appendix C.2 along with the same experiment over different languages. Across all five benchmark tasks our proposed ability estimator $\gamma_m^{\{\mathrm{mp,gmp}\}-\mathrm{IRT}}$ consistently yields ability vectors with higher cosine similarity to $\Gamma$ than $\gamma_m^{\{\mathrm{p,gp}\}-\mathrm{IRT}}$. This trend is evident across both subset sizes, highlighting the robustness of our approach even with limited data. The superior performance of $\gamma_m^{\{\mathrm{mp,gmp}\}-\mathrm{IRT}}$ empirically validates Assumption 1, confirming that an IRT-based ability estimator designed around this assumption provides more accurate ability estimates than a general-purpose alternative.

**Performance Estimators.** To assess the accuracy of our proposed performance estimators, we measure their absolute estimation error across different sample sizes. Specifically, we evaluate the performance estimates of six merged models using random sampling, P-IRT, GP-IRT (Polo et al., 2024), MP-IRT, and GMP-IRT across various subset sizes. The resulting absolute errors shown in Figure 3 are reported for `ARC`, `GSM8K`, `TruthfulQA`, and an aggregate average across all five benchmarks.

As shown in the figure, our proposed estimators, MP-IRT and GMP-IRT, consistently achieve lower absolute error compared to GP-IRT and P-IRT. While all IRT-based

methods outperform random sampling, the incorporation of merged-performance IRT significantly enhances estimation accuracy. Notably, both MP-IRT and GMP-IRT maintain low empirical error and reduced variance even when operating with very small subsets ($|\bar{D}| \approx 1.5\%$ of the full dataset). This highlights the robustness of our approach in low-data regimes.

Since lower empirical error often correlates with reduced *expected* error (as formalized in Section 5), we adopt MP-IRT and GMP-IRT as our primary estimators for evolving merged language models in subsequent experiments.

### 4.2. Cross-Lingual Transfer of Mathematical Skills

To assess the transfer of mathematical reasoning from English to other languages, we merge an English math-specialized model with a Mistral-7B (Jiang et al., 2023) fine-tuned on each target language, then evaluate on the corresponding GSM8K translations (Cobbe et al., 2021). Appendix B.2 provides details on the specific models used for merging. Following Akiba et al. (2025), we label an answer correct only if it is both accurate and written in the target language. We benchmark our approach against three commonly used merging baselines – Task Arithmetic (Ilharco et al., 2022), TIES (Yadav et al., 2023) and DARE (Yu et al., 2024). Following standard practice in the merging community, we apply either TIES and DARE jointly or SLERP (Shoemake, 1985).

As shown in fig. 5, merging a language-specific fine-tuning with a math-specialized model consistently surpasses both endpoint models by 10–20% in accuracy on the translated GSM8K. In contrast, standard baselines often yield sub-optimal merges, performing worse than the endpoints themselves. This highlights the importance of optimized merging coefficients and motivates our evolutionary framework.

Next, we evaluate our method for transferring math skills from English to Japanese and compare it to EvoMerge (Akiba et al., 2025), which serves as an upper bound by computing fitness on the full dataset. As illustrated in figure 6, our approach confirms the significant gains seen for the other languages, greatly surpassing both the performance of the endpoint models and that of the merging baselines. While the accuracy is lower than that of the model obtained by computing the fitness on the full dataset as done by Akiba et al. (2025), figure 1 shows that our approximation yields a method that is $50\times$ more efficient, effectively making evolutionary merging feasible on a single consumer GPU.

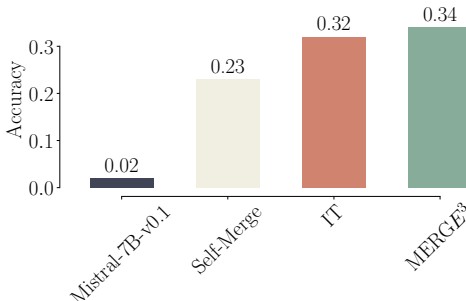

*Figure 7.* Accuracy of the base model (Mistral-7B), the Italian Endpoint (IT), Self-Merge and MERGE[3] models on the Italian-translated version of GSM8k.

### 4.3. Ablation Study: Self-Merging

In this section, we present an ablation study to test whether the observed improvements in the merged models arise from genuine cross-lingual knowledge transfer or merely from fitting to the prompt template. To structure this analysis, we formalize our inquiry through two hypotheses:

*Null Hypothesis ($H_0$).* The improvements seen in the merged models are due to the model fitting itself on the prompt template, rather than any cross-lingual knowledge exchange.

*Alternative Hypothesis ($H_1$).* The improvements arise from actual cross-lingual knowledge transfer and are not merely the result of fitting the prompt template.

To evaluate these hypotheses, we propose a *self-merging* procedure. Concretely, we take the linguistic model and merge it with *itself* using the standard MERGE[3] methodology outlined in algorithm 1. Under $H_0$, if the improvements are solely due to the prompt template, merging the model with itself should lead to performance gains (i.e., the merged model would still "fit" the template). Conversely, under $H_1$, if cross-lingual knowledge transfer is responsible for the enhanced performance, self-merging should *not* yield improvements. In fact, additional noise could even degrade performance relative to the baseline.

We conducted this self-merging experiment on the Italian model using the GSM8K dataset. The results, shown in fig. 7, reveal that performance actually *decreases* when the model is merged with itself. This observation strongly supports the alternative hypothesis ($H_1$): the performance gains in cross-lingual merges indeed stem from genuine knowledge transfer, rather than mere adaptation to a prompt template.

### 4.4. Evolving a Multilingual model

We next combine individually fine-tuned models for {IT, EN, DE, NL} into a single multilingual model. Ap-

*Table 1.* Evolving a multilingual model. For each language, we report the accuracy on the corresponding translated ARC of both the language-specific model and the evolved multilingual model.

| Model | Accuracy (↑) | | | | | | | |
|---|---|---|---|---|---|---|---|---|
| | Italian | | English | | German | | Dutch | |
| Finetuned | 0.61 | – | 0.75 | – | 0.61 | – | 0.50 | – |
| MERGE[3] | **0.69** | (↑8%) | **0.79** | (↑4%) | **0.72** | (↑11%) | **0.69** | (↑19%) |

pendix B.2 provides details on the specific models used for each language. As shown in table 1, the resulting merged model surpasses each language-specific variant by up to 19% in accuracy on the ARC-Challenge dataset (Clark et al., 2018). Even more notably, it outperforms all its constituent endpoints, demonstrating a clear positive transfer of knowledge across languages. Beyond the clear accuracy boosts in each language, a few key insights stand out. First, the largest improvement occurs for Dutch (from 50% to 69%), suggesting that merging particularly benefits languages where the baseline performance is lower. Second, even English, which starts from the highest baseline, still gains by 4%, indicating that positive transfer is not limited to low-resource or weaker endpoints. Finally, the fact that the merged model outperforms all individual fine-tunings (rather than landing between them) points to a genuine cross-lingual synergy, wherein knowledge from each language-specific model collectively strengthens the multilingual result. These conclusions are further strengthened by the ablation study in section 4.3, where we assess whether the observed improvements in the merged models arise from genuine cross-lingual knowledge transfer.

## 5. Theoretical Analysis

In this section, we provide theoretical guarantees for our *performance estimator*, demonstrating that its estimated accuracy is a reliable approximation of full-dataset accuracy. We provide formal guarantees for its performance, analyze its stability under dataset reduction, and explain why it remains a robust proxy for the true fitness of the merged models. This analysis not only solidifies the estimator's theoretical foundation but also offers practical insights into its behavior in finite-data and asymptotic regimes.

The section is structured as follows: first (§5.1), we derive a correlation between the accuracy of the performance estimator and the quality of the minimum found by solving an optimization problem using that performance estimator as objective function; second (§5.2), we study the asymptotic properties of the performance estimator as the dataset size approaches infinity, formalizing it as an unbiased estimator; and finally (§5.3), we demonstrate that our performance estimator behaves in expectation within a $\epsilon$-bound

of the accuracy on the true optimum dataset. The proofs for all the theorems and propositions presented below are outlined in appendix D.

### 5.1. Part I: $\epsilon$-Stable Estimators and $\epsilon$-Optimality Preservation

We first consider a performance metric $F(\theta; D)$ for $\theta \in \Theta \subset \mathbb{R}^n$, where $D$ is a dataset. If we choose a smaller subset $\bar{D} \subset D$ to approximate this metric, denoted $F(\theta; \bar{D})$, we wish to control the loss in optimality incurred by replacing $F(\theta; D)$ with $F(\theta; \bar{D})$.

**Definition 1** ($\epsilon$-Stability.)**.** Given two datasets $D$ and $\bar{D}$, we say $F(\cdot; \bar{D})$ is *$\epsilon$-stable with respect to* $F(\cdot; D)$ if, for all $\theta \in \Theta$,

$$\big| F(\theta; D) \ - \ F(\theta; \bar{D}) \big| \ \leq \ \epsilon$$

Under this condition, minimizing $F(\cdot; \bar{D})$ yields an objective value within $\epsilon$ of minimizing $F(\cdot; D)$. Formally:

**Theorem 2** ($\epsilon$-Optimality Preservation)**.** *Let $D$ be a dataset, let $\bar{D} \subset D$ be a subset, and let $F(\cdot; \bar{D})$ be $\epsilon$-stable with respect to $F(\cdot; D)$, with a fixed $\epsilon > 0$. Define*

$$\theta^\star \ = \ \underset{\theta \in \Theta}{\arg\min} \ F(\theta; D) \quad and \quad \hat{\theta} \ = \ \underset{\theta \in \Theta}{\arg\min} \ F(\theta; \bar{D})$$

*Then*

$$\big| F(\theta^\star; D) \ - \ F(\hat{\theta}; \bar{D}) \big| \ \leq \ \epsilon$$

Thus, $\epsilon$-stability ensures that any global minimizer on $\bar{D}$ achieves an objective value on $D$ no worse than $\epsilon$ from the true global optimum. Nevertheless, uniformly bounding $\big| F(\theta; D) - F(\theta; \bar{D}) \big|$ for all $\theta$ may be too strong in practice. For this reason, we introduce:

**Definition 3** ($\epsilon$-Stability in expectation)**.** Given two datasets $D$ and $\bar{D}$, we say $F(\cdot; \bar{D})$ is *$\epsilon$-stable in expectation with respect to* $F(\cdot; D)$ if

$$\mathbb{E}_{\bar{D}} \big[ \big| F(\theta; D) \ - \ F(\theta; \bar{D}) \big| \big] \ \leq \ \epsilon$$

where the expectation is over the (random) choice of $\bar{D}$

Under this relaxed notion, we still obtain a similar control on the *expected* suboptimality gap:

**Theorem 4** (Expected $\epsilon$-Stability of the Minimum)**.** *Suppose $F(\cdot; \bar{D})$ is $\epsilon$-stable in expectation with respect to $F(\cdot; D)$. Let*

$$m^\star \ := \ \min_{\theta \in \Theta} F(\theta; D) \quad and \quad \widehat{m}(\bar{D}) \ := \ \min_{\theta \in \Theta} F(\theta; \bar{D})$$

*Then*

$$\big| m^\star \ - \ \mathbb{E}_{\bar{D}} \big[ \widehat{m}(\bar{D}) \big] \big| \ \leq \ \epsilon$$

Hence, even if stability only holds *on average*, the expected gap between the global optimum on $D$ and the optimum on $\bar{D}$ remains at most $\epsilon$.

## 5.2. Part II: Theoretical Guarantees for MP-IRT

We now apply these ideas to our proposed MP-IRT estimator (cf. §3.1). We first show that MP-IRT is asymptotically unbiased, and then combine this fact with Theorem 4 to argue that MP-IRT-based minimizers remain close to those that minimize the full-dataset performance measure.

**Asymptotic unbiasedness.** The following proposition establishes that, as $\bar{D}$ grows, $\hat{Z}^{\text{mp-IRT}}$ converges in probability to the true performance $Z$. Its proof relies on classical limit arguments for unbiased estimators.

**Proposition 5** (Asymptotic unbiasedness of MP-IRT). Assume: (i) $\hat{\lambda} \to \lambda$ in probability as $|\hat{I}| \to \infty$, (ii) for each $i \in I$, the true values $\alpha_i, \beta_i, \theta_1, \theta_2$ are known, with $\sup_{i \in I} \|\alpha_i\|_2 \leq c$ for a fixed $c$, (iii) linear inheritance of abilities (cf. Assumption 1) holds. Then, for all $j, l$,

$$\left| \mathbb{E}\big[\hat{Z}_{jl} \mid Y_{i_0 l}, \ldots, Y_{i_k l}\big] - \mathbb{E}\big[Z_{jl} \mid Y_{i_0 l}, \ldots, Y_{i_k l}\big] \right| \to 0$$

in probability as $|\hat{I}| \to \infty$. Thus, for sufficiently large subsets $\bar{D}$, the discrepancy between $\hat{Z}_{\tilde{m}}$ and $Z_{\tilde{m}}$ can be made arbitrarily small with high probability.

## 5.3. Part III: performance preservation via MP-IRT

We now conclude that MP-IRT preserves near-optimality when we train on a suitably large $\bar{D} \subset D$. Since Proposition 5 asserts that $\hat{Z}$ approximates $Z$ well for large $|\bar{D}|$, it follows (under mild conditions) that MP-IRT remains $\epsilon$-stable in expectation. Hence, Theorem 4 shows that minimizing $\hat{Z}$ on $\bar{D}$ yields, on average, a solution within $\epsilon$ of the full-dataset optimum.

**Theorem 6** (Asymptotic performance preservation of MP-IRT). *Let $\bar{D} \subset D$ be a random subset used to compute $\hat{Z}^{\text{mp-IRT}}$. Suppose that, as $|\bar{D}| \to \infty$, $\hat{Z}^{\text{mp-IRT}}$ converges in probability to $Z$ (the true performance on $D$), and that $\hat{Z}^{\text{mp-IRT}}$ is $\epsilon$-stable in expectation for sufficiently large $|\bar{D}|$. Then the expected global optimum of $\hat{Z}^{\text{mp-IRT}}$ on $\bar{D}$ differs from that of $Z$ on $D$ by at most $\epsilon$. As $|\bar{D}| \to \infty$, $\epsilon \to 0$.*

**Finite-sample analysis via the Law of Large Numbers.** In practice, we rarely have $|\bar{D}| \to \infty$. Instead, one can appeal to *expected $\epsilon$-stability* (Theorem 4) and then *estimate* the corresponding expectation empirically. For instance, one may draw multiple subsets $\bar{D}_1, \ldots, \bar{D}_S$ at random from $D$ and compute

$$\frac{1}{S} \sum_{s=1}^{S} \left| F(\theta; D) - F(\theta; \bar{D}_s) \right|$$

as an empirical approximation to $\mathbb{E}_{\bar{D}}\big[|F(\theta; D) - F(\theta; \bar{D})|\big]$. By the Law of Large Numbers, if this empirical

average remains small (say, $\approx \tilde{\epsilon}$), then the true expectation is also small. Consequently, Theorem 4 implies that the optimal solution on each $\bar{D}_s$ is within $\tilde{\epsilon}$ of the global optimum on $D$, on average.

**Conclusion.** In summary, MP-IRT inherits asymptotic consistency from p-IRT while requiring only a subset $\bar{D} \subset D$. By showing it is $\epsilon$-stable (in expectation) for large $|\bar{D}|$, we conclude that *optimizing on $\bar{D}$ yields (on average) a solution close to the true optimum on $D$*. In finite-sample regimes, multiple random draws of $\bar{D}$ can be used to empirically verify that the discrepancy remains small, thereby justifying the practical use of MP-IRT on moderately sized subsets.

## 6. Technical Details

We summarize the GPU timing results for MERGE$^3$ in table 2, comparing evaluation and merge times across different hardware setups. These findings highlight the practical feasibility of our approach even on older GPUs. For additional experimental details refer to appendix C.3.3.

## 7. Conclusions

We introduced MERGE$^3$, an evolutionary merging framework that makes high-quality model merging feasible on a single consumer GPU. By combining a subset-based approach with IRT-driven performance estimation, MERGE$^3$ reduces merging costs by up to fifty-fold compared to prior methods – without sacrificing the quality of the merged model. Our experiments demonstrate successful cross-lingual transfer in mathematics (e.g., from English to Japanese), as well as the synthesis of new multilingual models that outperform each of their language-specific endpoints. Overall, MERGE$^3$ expands the practical reach of evolutionary merging, allowing everyday practitioners to benefit from advanced multi-task and multilingual model compositions at a fraction of the usual computational cost.

*Table 2.* Comparison of Evolve methods by number of trials, estimated total time on a single NVIDIA 4090, sample size used for Fitness computation, and final accuracy on GSM8K. The number of trials is the result of population size × iterations, parameters of the Genetic Algorithms of each method, and represents the total number of merged models evaluated during the entire Evolve run.

| Method | $N_{\text{models}}$ | Estimated total time | Sample size | Accuracy |
|---|---|---|---|---|
| EvoLLM-JP-7B | 1000 | 62 days | 1000 | 0.49 |
| MERGE$^3_{100}$ | 175 | 21h | 100 | 0.42 |
| MERGE$^3_{50}$ | 175 | 12h 20m | 50 | 0.38 |
| MERGE$^3_{30}$ | 175 | 10h 30m | 30 | 0.38 |
| MERGE$^3_{20}$ | 175 | 10h 15m | 20 | 0.34 |

## Impact Statement

The introduction of MERGE[3] provides an efficient and accessible method for evolutionary model merging on consumer-grade GPUs. By combining dataset reduction techniques and Item Response Theory (IRT)-based performance estimations, MERGE[3] significantly lowers computational requirements while maintaining competitive performance. This enables researchers and developers to synthesize high-quality multilingual and cross-lingual models without requiring cluster-scale hardware.

The open-source release of MERGE[3] aims to make evolutionary model merging widely accessible, fostering further innovation in resource-constrained environments. With applications in multilingual NLP and low-resource language modeling, MERGE[3] addresses practical challenges in the field, offering an efficient solution for creating state-of-the-art models on standard hardware.

ACKNOWLEDGMENTS

We acknowledge ISCRA for awarding this project access to the LEONARDO supercomputer, owned by the EuroHPC Joint Undertaking, hosted by CINECA (Italy). We also acknowledge support from Sapienza University of Rome, under project BEAT (Better dEep leArning securiTy) and through the Seed of ERC grant "MINT.AI".

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

*Table 3.* Overview of supported merging methods in Mergenetic.

| Method | Multi-Model | Uses Base Model |
|---|---|---|
| Linear (Model Soups) | Yes | No |
| SLERP | No | Yes |
| Task Arithmetic | Yes | Yes |
| TIES | Yes | Yes |
| DARE (TIES) | Yes | Yes |
| DARE (Task Arithmetic) | Yes | Yes |

## A. Mergenetic

Each experiment was run using a library developed specifically for this paper, which will be released as open-source software, called *Mergenetic* (Minut et al., 2025). This library allows for defining a merging problem as either a single-objective or multi-objective optimization problem, where one only needs to specify the merging method, a fitness function, and select the hyperparameters for a chosen evolutionary algorithm.

The implementation relies on *Mergekit* (Goddard et al., 2024) for merging the models, *Pymoo* (Blank & Deb, 2020) for optimizing the objective function through evolutionary algorithms, and *Lm-Evaluation-Harness* (Gao et al., 2024) for implementing some of the fitness functions. In table 3 we outline the supported merging methods, while in table 4 we outline the currently available evolutionary algorithms.

We believe this library is a significant contribution as it facilitates evolutionary model merging and aligns well with the paper's approach, which aims to reduce computational burden. It can be a valuable tool for the community and for users interested in cross-lingual transfer or creating multi-lingual models for target low-resource languages.

## B. Additional Details

This section provides additional implementation and experimental details that were not included in the main paper.

### B.1. IRT Fitting Details

As previously stated, we used the implementation from Polo et al. (2024) and adopted their configuration settings. Specifically, we used $\gamma_m \sim N(\mu_\gamma \mathbf{1}_d, 1/u_\gamma I_d)$, $\alpha_i \sim N(\mu_\alpha \mathbf{1}_d, 1/u_\alpha I_d)$, and $\beta_i \sim N(\mu_\beta, 1/u_\beta)$. Following Polo et al. (2024), we also applied (hyper)priors to the prior parameters using the software for fitting hierarchical Bayesian models (Lalor & Rodriguez, 2023): $\mu_\gamma \sim N(0, 10)$, $u_\gamma \sim \Gamma(1, 1)$, $\mu_\alpha \sim N(0, 10)$, $u_\alpha \sim \Gamma(1, 1)$, $\mu_\beta \sim N(0, 10)$, and $u_\beta \sim \Gamma(1, 1)$. For both the model and example-specific parameters $\gamma_m$, $\alpha_i$, and $\beta_i$, we take

*Table 4.* Overview of supported Pymoo's evolutionary algorithms in Mergenetic.

| Algorithm | Class | Objective(s) | Constraints |
|---|---|---|---|
| Genetic Algorithm | GA | single | x |
| Differential Evolution | DE | single | x |
| Biased Random Key GA | BRKGA | single | x |
| Nelder Mead | NelderMead | single | x |
| Pattern Search | PatternSearch | single | x |
| CMAES | CMAES | single | |
| Evolutionary Strategy | ES | single | |
| SRES | SRES | single | x |
| ISRES | ISRES | single | x |
| NSGA-II | NSGA2 | multi | x |
| R-NSGA-II | RNSGA2 | multi | x |
| NSGA-III | NSGA3 | many | x |
| U-NSGA-III | UNSGA3 | many | x |
| R-NSGA-III | RNSGA3 | many | x |
| MOEAD | MOEAD | many | |
| AGE-MOEA | AGEMOEA | many | |
| C-TAEA | CTAEA | many | x |
| SMS-EMOA | SMS-EMOA | many | x |
| RVEA | RVEA | many | x |

*Table 5.* Mistral-based models with shortened column headers and names. Role can be either E, M or B, referring to endpoint, merge or base model respectively. Spec refers instead to specialization, with `mth`, `ger`, `ita`, `jpn`, `dut` and `gen` referring to Math, German, Italian, Japanese, Dutch and General respectively. We finally have the author and model ID as per the Huggingface.

| Role | Spec | Author | Model |
|---|---|---|---|
| E | `mth` | *upaya07* | `Arithmo2-Mistral-7B` |
| E | `mth`, `jpn` | *SakanaAI* | `EvoLLM-JP-v1-7B` |
| E | `mth` | *GAIR* | `Abel-7B-002` |
| E | `mth` | *meta-math* | `MetaMath-Mistral-7B` |
| B | `gen` | *mistralai* | `Mistral-7B-v0.1` |
| E | `ger` | *jphme* | `em_german_mistral_v01` |
| E | `ger` | *LeoLM* | `leo-mistral-hessianai-7b` |
| E | `ita` | *DeepMount00* | `Mistral-Ita-7b` |
| E | `jpn` | *augmxnt* | `shisa-gamma-7b-v1` |
| E | `dut` | *BramVanroy* | `GEITje-7B-ultra` |
| E | `ro` | *OpenLLM-Ro* | `RoMistral-7b-Instruct` |
| M | `gen` | *chlee10* | `T3Q-Merge-Mistral7B` |
| E | `gen` | *liminerity* | `M7-7b` |
| E | `gen` | *yam-peleg* | `Experiment26-7B` |
| M | `gen` | *PracticeLLM* | `SOLAR-tail-10.7B-Merge-v1.0` |
| E | `gen` | *upstage* | `SOLAR-10.7B-v1.0` |
| E | `gen` | *Yhyu13* | `LMCocktail-10.7B-v1` |
| M | `gen` | *FuseAI* | `FuseChat-7B-Slerp` |
| M | `gen` | *FuseAI* | `FuseChat-7B-TA` |
| E | `gen` | *FuseAI* | `OpenChat-3.5-7B-Mixtral` |
| E | `gen` | *FuseAI* | `OpenChat-3.5-7B-Solar` |
| M | `gen` | *jan-hq* | `supermario-slerp-v3` |
| E | `gen` | *jan-hq* | `supermario-slerp-v2` |
| E | `gen` | *jan-hq* | `supermario-v2` |
| M | `gen` | *superlazycoder* | `NeuralPipe-7B-slerp` |
| E | `gen` | *OpenPipe* | `mistral-ft-optimized-1218` |
| E | `gen` | *mlabonne* | `NeuralHermes-2.5-Mistral-7B` |

their point estimates as the means of their respective variational distributions. The $\gamma$ model dimensionality is set to 15 following the parameter choice suggested by Polo et al. (2024).

## B.2. Experimental Details

**Models.** One key assumption of model merging is that the endpoint models lie within the same basin (Ilharco et al., 2022). This means that merging arbitrary models is not feasible; rather, **all models involved must be fine-tuned versions of the same base model**. To satisfy this requirement, we selected several fine-tuned models from the Hugging Face Hub that originated from the same base model. Specifically, we focused on models fine-tuned starting from `Mistral-7B` (Jiang et al., 2023), following common best practices in the community (Akiba et al., 2025). Table 5 lists all the models used in our experiments, along with their corresponding names on the Hugging Face Hub. A total of 27 models were considered for our experiments.

**Number of models.** The initialization step, shared across all evolutionary algorithms, involves sampling merging configurations (e.g., interpolation coefficients) and applying them to merge the endpoint models. Consequently, MERGE[3] requires the same number of models as any standard merging approach.

In the rest of the section, we provide further details for reproducing the experiments in section 4.2 and section 4.4 of the main paper.

### B.2.1. CROSS-LINGUAL TRANSFER

In the cross-lingual transfer evolutionary merging, we evolved four merged models with mathematical capabilities in different languages: Japanese, Romanian, German, and Dutch. In each of these experiments, we deployed an ad-hoc genetic algorithm for single-objective optimization. We employed the Simulated Binary Crossover (Deb et al., 2007) operator to generate offspring solutions by combining parent solutions. To maintain diversity and explore the search space, we applied Polynomial Mutation (Deb et al., 2007), which introduces small perturbations to offspring solutions and enhances the algorithm's ability to escape local optima. This combination of SBX and PM effectively balances exploration and exploitation, facilitating efficient convergence toward optimal solutions.

Furthermore, guided by empirical tests, we decided to deploy `SLERP` to evolve solutions for the Romanian and Dutch problems, while we used a combination of `TIES` and `DARE` for the Japanese and the German ones. We deployed four different sizes of the fitness datasets for Japanese, namely 20, 30, 50, and 100, in order to obtain a more detailed analysis of the method for comparison with the work of (Akiba et al., 2025). On the other hand, we kept the fitness dataset size fixed to 20 for all other aforemen-

tioned experiments. The fitness dataset was extracted from the test set of GSM8K, and we used the remaining, non-overlapping samples as test set for evaluating the model. To get the language-specific versions of GSM8K, we used Unbabel/TowerInstruct-7B-v0.2 (Alves et al., 2024) to translate the datasets. In each experiment, the population size was fixed to 25 and the number of iterations to 7.

To check the correctness of the solution, following Akiba et al. (2025), we used a regex to extract the last numerical value returned in the model's answer and compare it with the ground truth. The solution is also checked to be in the correct language with the language identifier from fastText (Joulin et al., 2017).

The mathematical models used in combination with TIES and DARE were Abel-7B-002 and Arithmo2-Mistral-7B, whereas we used MetaMath-Mistral-7B in combination with SLERP. Moreover, we employed the following language-specialized models: shisa-gamma-7b-v1, em_german_mistral_v01, GEITje-7B-ultra, and RoMistral-7b-Instruct. More information about these models can be found in table 5.

Lastly, we evaluated EvoLLM-JP-v1-7B (Akiba et al., 2025) under the same conditions as MERGE[3] to assess its accuracy, following the prompting structure outlined by Akiba et al. (2025).

### B.2.2. MULTI-LINGUAL TRANSFER

In this experiment, we tackle the ARC dataset in multiple languages (Italian, Dutch, German, and English)[2] (Thell-mann et al., 2024) using a multi-objective evolutionary merging procedure based this time on NSGA-II (Deb et al., 2002). We configure the population size to 25 and the number of evolutionary iterations to 7. We deployed a combination of TIES and DARE as merging strategy. As in previous settings, both the fitness function and the test metrics operate by extracting the final model-generated choice via a regex, but this time they look for an instance from the set {A, B, C, D} rather than a number. On top of this, we employed a dataset composed by 20 datapoints for each language from the relative translation of ARC to compute the fitness, and we extracted the test set as for the previous experiments. Furthermore, unlike the single-objective approach described earlier, here we explicitly optimize multiple objectives simultaneously. This time, the employed models are Mistral-Ita-7b, GEITje-7B-ultra, leo-mistral-hessianai-7b, and the base model Mistral-7B-v0.1.

---

[2]We used the dataset on the Hugging Face Hub from openGPT-X/arcx

*Table 6.* Notation used in the paper.

| Notation | Description |
|---|---|
| $D$ | Full dataset. |
| $\bar{D}$ | Reduced subset of the dataset. |
| $D_i$ | Subdataset for task $i$. |
| $\gamma_m$ | Latent abilities of model $m$. |
| $\Gamma_m$ | True latent abilities of model $m$. |
| $\gamma_m^{\{p,gp\}-IRT}$ | Latent abilities of model $m$ via P-IRT ability estimator. |
| $\gamma_m^{\{mp,gmp\}-IRT}$ | Latent abilities of model $m$ via MP-IRT ability estimator. |
| $\alpha_i, \beta_i$ | IRT parameters related to dataset item $i$. |
| $\lambda$ | Interpolation coefficients for latent abilities. |
| $\hat{\lambda}, \hat{\gamma}, \hat{\alpha}, \hat{\beta}$ | MLE of the aforementioned parameters. |
| $p_{i,m}$ | IRT model for datapoint $i$ and model $m$. |
| $\hat{p}_{i,m}$ | IRT model for datapoint $i$ and model $m$ parametrized by MLE estimators of $\alpha, \beta, \gamma, \lambda$. |
| $\tilde{m}$ | Merged language model. |
| $Y_{i,m}$ | Sample-level correctness of model $m$ for example $i$. |
| $\hat{Z}^{\text{MP-IRT}}$ | Merged performance estimator MP-IRT. |
| $\hat{Z}^{\text{GMP-IRT}}$ | Generalized merged performance estimator GMP-IRT. |
| $F(m)$ | Fitness value of a model $m$. |
| $\theta$ | Parameters being optimized in evolutionary search. |
| $P_{\bar{F}_D}$ | Pareto front defined by function's set $\bar{F}$ and data $D$ |
| $\theta^\star$ | Global optimum on $D$. |
| $\hat{\theta}$ | Global optimum on $\bar{D}$. |
| $N$ | Number of samples in the dataset. |

### B.2.3. ABILITY AND PERFORMANCE ESTIMATOR

In these experiments (reported in section 4.1 and section 4.1) we used the test set of the standard version of GSM8K, HellaSwag, ARC, Winogrande, and TruthfulQA. Furthermore, we used 6 different models to test the different performance of the ability and performance estimator: SOLAR-tail-10.7B-Merge-v1.0, FuseChat-7B-Slerp, NeuralPipe-7B-slerp, T3Q-Merge-Mistral7B, FuseChat-7B-TA, and supermario-slerp-v3. These models were chosen as already available on the Open LLM Leaderboard.

## C. Additional Experiments

We report here additional experiments and analyses.

### C.1. Extract Step

In the extract step outlined in section 3.1, random sampling has been proposed as the main method to subsample the dataset $\bar{D} \subset D$. While we explored various dataset sub-sampling strategies, we ultimately opted for uniform random sampling, as our experiments showed that more complex approaches offered no significant advantage over this simpler method. In this section we report some of the experiments behind this decision and the two alternative methods tried in the extraction step: IRT Clustering (IRT), introduced by Polo et al. (2024), and a custom Representation Clustering (RC) method.

### C.1.1. IRT CLUSTERING

Given a dataset $D$ and the parameter of a fitted IRT model $\alpha$ and $\beta$, one can define a low-dimensional embedding of each datapoint $i \in D$ by $E_i = [\alpha_i \| \beta_i]$. Therefore, IRT-clustering obtains a representative subset by first obtaining a clustering over this embedding space through $K$-Means, and then choosing the points closest to the centroids as representative samples.

### C.1.2. REPRESENTATION CLUSTERING

Let $\{m_j\}_{j=1}^M$ be the set of endpoint models, and let $D = \{x_i\}_{i=1}^N$ be our full dataset. For each sample $x_i$, we first encode it into a high-dimensional vector by concatenating model-specific embeddings. Concretely, we compute:

$$E_{i,j} = \frac{1}{T_i} \sum_{t=1}^{T_i} E_{i,j,t} \in \mathbb{R}^d,$$

where $E_{i,j,t}$ is the embedding of the $t$-th token of sample $x_i$ under model $m_j$, and $T_i$ is the number of tokens in $x_i$. We form the concatenated representation:

$$E_i = [E_{i,1} \| E_{i,2} \| \cdots \| E_{i,M}] \in \mathbb{R}^{M \cdot d}.$$

Since $E_i$ can be very high-dimensional, we apply Principal Component Analysis (PCA) to project $E_i$ onto a lower-dimensional space:

$$\tilde{E}_i = \text{PCA}_k(E_i) \in \mathbb{R}^k, \quad k \ll M \cdot d.$$

Next, we apply $k$-means clustering to the reduced embeddings $\{\tilde{E}_i\}_{i=1}^N$:

$$\min_{\{\mathbf{c}_k\}_{k=1}^K} \sum_{i=1}^N \min_{1 \le k \le K} \|\tilde{E}_i - \mathbf{c}_k\|^2,$$

where $\mathbf{c}_k$ is the centroid of the $k$-th cluster. This partitions the dataset into $K$ clusters, each capturing a distinct region of the representation space. From each cluster $k$, we select the representative sample $x_{i_k^\star}$ whose embedding $\tilde{E}_{i_k^\star}$ is closest to the centroid $\mathbf{c}_k$:

$$i_k^\star = \arg \min_{x_i \in C_k} \|\tilde{E}_i - \mathbf{c}_k\|,$$

where $C_k$ is the set of samples assigned to cluster $k$. To approximate the full-dataset metrics from the selected subset $\bar{D} = \{x_{i_k^\star}\}_{k=1}^K$, we assign a weight to each representative sample. Since the size of the cluster $C_k$ indicates how prevalent that region of representation space is, we define $w_{i_k^\star} = \frac{|C_k|}{|D|}$. These weights ensure that the contribution of each representative sample to the overall metric reflects the true proportion of samples that it represents in the original dataset. By evaluating a new model $m$ only on $\bar{D}$ and using $\{w_{i_k^\star}\}$ to calculate a weighted average, we approximate

$m$'s performance on the full dataset $D$ at a fraction of the computational cost.

A schematic overview of the full process is outlined in algorithm 2.

---

**Algorithm 2** Representation Clustering Extractor

---

**Require:** Dataset $D$, Endpoint Models $m_1, ..., m_n$, Desired subset size $K$
**Ensure:** Subset of size $K$ with weights $w_i$
1: **for** $i$ in $D$ **do**
2:    $E_i \leftarrow []$
3:    **for** $m$ in $\{m_1, \ldots, m_n\}$ **do**
4:       $E_{im} \leftarrow$ embed $i$ with model $m$
5:       $E_i \leftarrow E_i | E_{im}$
6:    **end for**
7: **end for**
8: $\{\mathbf{E}_i\}_{i \in D} \leftarrow \text{PCA}(\{\mathbf{E}_i\}_{i \in D})$
9: Apply k-means clustering to $\{\mathbf{E}_i\}_{i \in D}$, obtaining $K$ centroids $\{\mathbf{c}_k\}_{k=1}^K$
10: For each cluster $k$, select the closest example $i_k^\star = \arg \min_{i \in D} \|\mathbf{E}_i - \mathbf{c}_k\|_2$
11: Let $C_k = \{i \in D \mid \arg \min_{c \in \{\mathbf{c}_k\}_{k=1}^K} \|\mathbf{E}_i - c\|_2 = \mathbf{c}_k\}$ be the set of examples in cluster $k$
12: Assign weights $w_{i_k^\star} = \frac{|C_k|}{|D|}$ for $k = 1, ..., K$
13: **return** $\{i_k^\star, w_{i_k^\star}\}_{k=1}^K$

---

### C.1.3. EXPERIMENTS

To compare the performance of the Sample Extractors, we followed a procedure similar to that described in section 4.1, computing the absolute estimation error for each extractor. For random sampling, the accuracy estimator was obtained via uniform averaging, whereas for IRT and RC it was obtained via weighted averaging. We evaluated the estimator in two different settings: (1) merging a math model with a language-tuned model (similar to the cross-lingual setting of section 4.2) for several languages (Italian, German, Romanian, Dutch) and testing the extractor on the corresponding translations of GSM8K (see fig. 8), and (2) merging several math models and testing the extractor on the English version of GSM8K (see fig. 9).

Focusing on fig. 8, we see that performance variability is somewhat higher (larger error bars) due to different language-specific datasets. Even so, Random sampling never falls behind IRT or RC, especially for small sample sizes. By the time the subset size reaches 50 or more examples, all three methods converge to comparable accuracy-error levels, underscoring the robustness of Random sampling. Instead, in fig. 9, the trends are broadly similar for RC and Random sampling, while slightly worse for IRT. Again, as the dataset sample size grows, overall error drops and the gap among methods narrows.

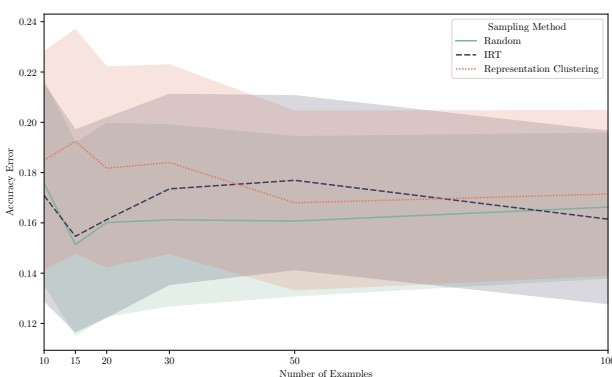

*Figure 8. Extractors across Languages:* Absolute error of the estimated accuracy of Sample Extractors, averaged across merges of language-specific and English Math finetunings of `Mistral-7B-v0.1`, evaluated on translations of `GSM8K` and presented as a function of the number of dataset samples.

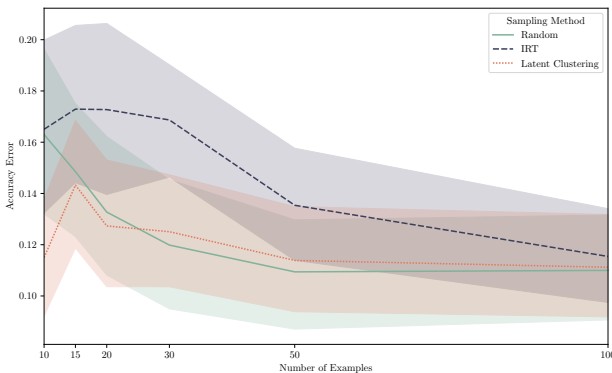

*Figure 9. Extractors across Merges:* Absolute error of the estimated accuracy of Sample Extractors, averaged across merges of English Math models based on `Mistral-7B-v0.1`, evaluated on `GSM8K` and presented as a function of the dataset sample size.

To sum up, the Random sampler can sometimes lag slightly behind the more sophisticated IRT and RC. Nevertheless, neither of these methods has a clear advantage over the others. Given its simplicity and negligible overhead, the Random strategy stands out as a highly practical choice for dataset subsampling—especially when the marginal improvements of more complex methods do not clearly justify their added complexity.

### C.2. Estimation step

#### C.2.1. ADDITIONAL EXPERIMENT FOR ABILITY ESTIMATOR

We report in fig. 10 the Euclidean distance between the estimated and ground-truth ability vectors across different sample sizes. The results are consistent with the case $n = 10, 20$ seen in fig. 4, with our estimated ability vec-

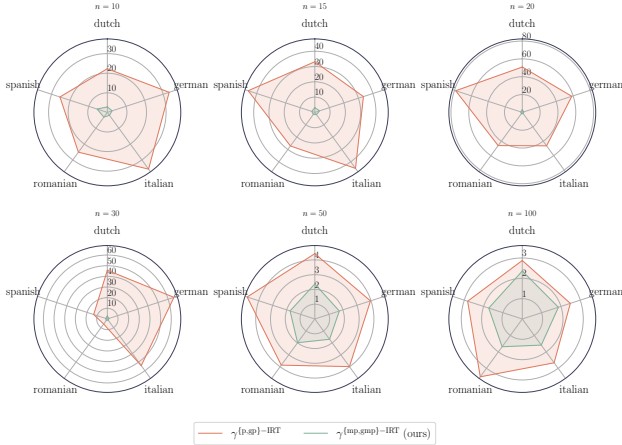

*Figure 10. Ability Estimator over languages:* Euclidean distance (lower is better) between estimated and true abilities for different languages.

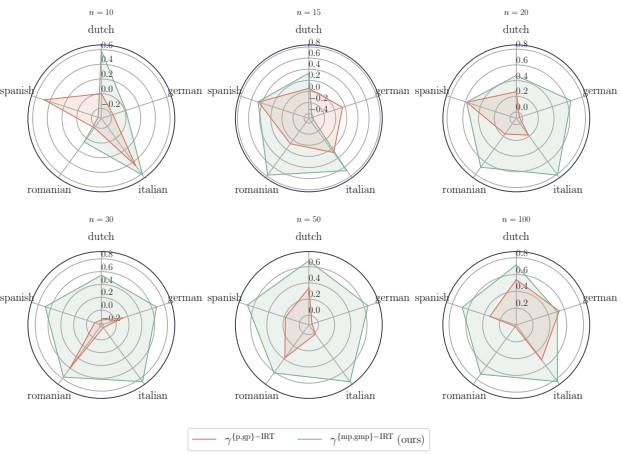

*Figure 11. Ability Estimator over languages:* Cosine similarity (higher is better) between estimated and true abilities for different languages.

tor being significantly closer to the ground-truth one compared to the ability vector estimated by pIRT and gp-IRT. Similarly, we report the corresponding cosine similarity in fig. 11, confirming much higher similarity in our case.

#### C.2.2. ADDITIONAL EXPERIMENT FOR PERFORMANCE ESTIMATOR

We report in fig. 13 the evaluation of performance estimators across `Winogrande` and `Hellaswag`, extending the results in fig. 3.

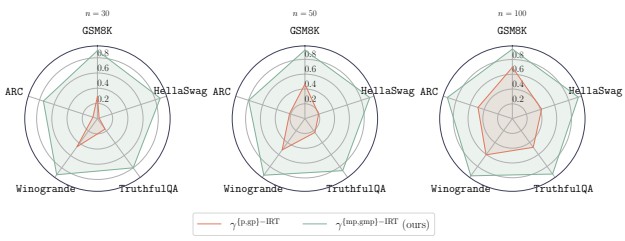

*Figure 12. Ability Estimator over tasks:* Cosine similarity (higher is better) between estimated and true abilities for different tasks.

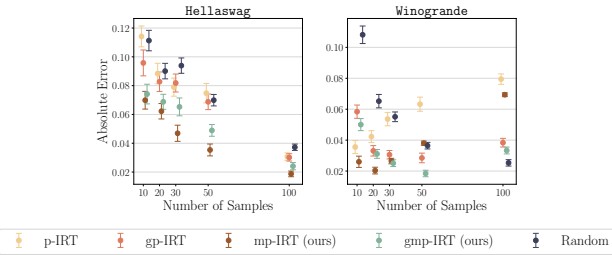

*Figure 13. Performance Estimators over Winogrande and Hellaswag.* Absolute error of various estimators as a function of sample size (lower is better). gmp-IRT consistently achieves lower error.

### C.2.3. HYPERPARAMETER ANALYSIS FOR PERFORMANCE ESTIMATOR

We now analyse the optimal choice of the scalar $c$ required in GMP-IRT and GP-IRT (see eq. (5)). In the experiments reported in the main paper (section 4.1) and above (appendix C.2.2), we used as a heuristic $c = \frac{1}{2}$. Despite its empirical effectiveness, this uniform interpolation may not be optimal across all model pairs and data regimes. Therefore, we introduce a grid-search-based strategy to estimate an improved value of $c$ and empirically validate its potential to reduce estimation error.

**Methodology.** We propose a two-step approach to selecting the optimal interpolation coefficient $c$ for use in the GMP-IRT estimator. The procedure is as follows:

1. **Optimizing $c$ for Endpoint Models** For a given dataset $\bar{D}$ [3], we find a $c \in [0, 1]$ by minimizing the absolute estimation error of GP-IRT on each individual model. This yields a set of optimal coefficients $\{c_1, \ldots, c_M\}$, one for each endpoint model $M$.

2. **Averaging $c$ for the Merged Model.** To obtain a suitable interpolation coefficient for GMP-IRT, we compute the average of the optimal values across the end-

---

[3]Such a dataset is available because it relies solely on the correctness of the endpoint models' answers, rather than on the unknown answers of the merged model.

point models, $\bar{c} = \frac{1}{M} \sum_{m=1}^{M} c_m$, and use this as the parameter for the merged model's GMP-IRT.

**Results Discussion** We evaluate the absolute estimation error across five benchmarks, comparing the adaptive averaging strategy for $c$ (denoted by $\star$) to the baseline fixed heuristic $c = \frac{1}{2}$. Results for GMP-IRT$^\star$ and GP-IRT$^\star$, along with the baseline models GMP-IRT, GP-IRT, MP-IRT, and P-IRT, are reported in table 7.

*Table 7.* Absolute estimation error across datasets. The $\star$ symbol indicates the adaptive $c$ strategy described above. Lower is better.

| **Dataset** | **GMP-IRT$^\star$** | **GMP-IRT** | **GP-IRT$^\star$** | **GP-IRT** | **MP-IRT** |
|---|---|---|---|---|---|
| ARC | **0.035** | 0.040 | 0.046 | 0.049 | 0.048 |
| Winogrande | **0.018** | 0.031 | 0.032 | 0.037 | 0.036 |
| GSM8K | **0.057** | **0.057** | 0.074 | 0.064 | 0.062 |
| HellaSwag | **0.046** | 0.056 | 0.077 | 0.071 | 0.047 |
| TruthfulQA | **0.040** | 0.045 | 0.062 | 0.055 | 0.044 |

Across most datasets, the adaptive coefficient strategy yields consistent improvements in both GMP-IRT and GP-IRT, with the largest gains observed on Winogrande and HellaSwag. Notably, GMP-IRT with a tuned $c$ performs on par or better than any other method across all benchmarks.

### C.2.4. SPEARMAN CORRELATION

Following White et al. (2021), we report the Spearman correlation between the ground-truth ranking and the ranking induced by each estimator's predictions. We compare the best merging-specific estimator, GMP-IRT, against the strongest vanilla baseline, GP-IRT. The results, averaged across dataset sizes and types, are shown in fig. 14. Notably, GMP-IRT achieves the highest correlation in each setting, further underscoring the benefits of using estimators specifically designed for the model merging context.

## C.3. Evolve Step

### C.3.1. ADDITIONAL EXPERIMENT FOR MULTILINGUAL EVOLUTION: COMPARISON WITH IN-CONTEXT LEARNING

A natural question when evaluating merging-based methods is how their performance compares to inference-time adaptation strategies such as *In-Context Learning* (ICL). In particular, given access to a small validation set, one might ask whether directly providing these examples as input context at evaluation time can match the performance achieved through evolutionary merging.

To investigate this, we evaluate a 20-shot ICL setup in the multilingual transfer setting introduced in section 4.4. Prompts are constructed using 20 validation examples and prepended to the input at inference time. We apply this setup to two merging baselines, TIES-DARE and Task

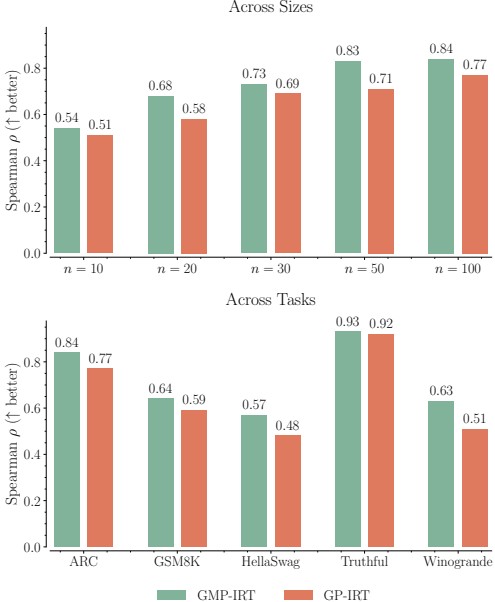

*Figure 14.* Spearman rank correlations using the appendix C.2.3 setup. Higher is better. GMP-IRT shows substantially higher correlation than GP-IRT.

Arithmetic, and compare the results against our method using GMP-IRT with a fitness dataset of size 20. Across all

*Table 8.* Accuracy on multilingual ARC using few-shot ICL (20-shot) versus MERGE[3] (GMP-IRT-20).

| Method | DE | IT | NL | EN |
|---|---|---|---|---|
| TIES-DARE Few-shot (20) | 0.227 | 0.226 | 0.227 | 0.226 |
| Task Arithmetic Few-shot (20) | 0.427 | 0.406 | 0.491 | 0.566 |
| MERGE[3] (GMP-IRT-20) | **0.720** | **0.690** | **0.690** | **0.790** |

languages, MERGE[3] significantly outperforms the ICL-augmented baselines. While ICL can provide moderate improvements over the original models, it increases inference-time memory usage and latency due to the expanded context. In contrast, the merged models produced by our method operate without additional overhead and are immediately deployable as standalone networks.

### C.3.2. ADDITIONAL EXPERIMENT FOR CROSS-LINGUAL EVOLUTION: ANALYZING NEGATIVE TRANSFER

While our main analysis focused on cross-lingual transfer of MERGE[3] (section 4.2), we did not explicitly examine the phenomenon of *negative transfer*; that is, cases where merging degrades performance on specific inputs. In this subsection, we formally define negative transfer in the context of multiple-choice questions (MCQs) and introduce a

framework for measuring its prevalence in merged multilingual models. We then present an analysis of negative transfer in GSM8K.

**Methodology** We consider a multiple-choice question (MCQ) evaluation setup, where knowledge is operationalized as a model's ability to correctly answer a question. Let $m_1, m_2, \ldots, m_K$ denote the set of $K$ endpoint models, and let $\tilde{m}$ represent the merged model resulting from their combination. Following our earlier notation (see table 6), correctness for a given sample $i$ is defined as a binary variable:

- $Y_{i,m_j} \in \{0, 1\}$: indicates whether endpoint model $m_j$ answers sample $i$ correctly,

- $Y_{i,\tilde{m}} \in \{0, 1\}$: indicates whether the merged model answers sample $i$ correctly.

We define negative transfer on example $i$ as occurring when at least one of the base models answers correctly, but the merged model fails:

$$\exists j \in \{1, \ldots, K\} \text{ such that } Y_{i,m_j} = 1 \quad \text{and} \quad Y_{i,\tilde{m}} = 0.$$

To track this, we introduce a binary indicator variable $n_i$ for each input:

$$n_i = \begin{cases} 1, & \text{if } \left(\exists j : Y_{i,m_j} = 1\right) \text{ and } Y_{i,\tilde{m}} = 0, \\ 0, & \text{otherwise.} \end{cases}$$

Finally, we compute the *Negative Transfer Rate (NTR)* as the proportion of examples exhibiting negative transfer among those for which at least one base model answered correctly:

$$\text{NTR} = \frac{\sum_{i=1}^{N} n_i}{\sum_{i=1}^{N} \mathbf{1}\left\{\exists j \in \{1, \ldots, K\} : Y_{i,m_j} = 1\right\}}.$$

This metric provides a task-level perspective on the potential degradation introduced by merging and complements the aggregate performance measures reported in the main paper.

**Results Discussion** We compute the NTR using the same experimental setting described in section 4.2. As shown in fig. 15, MERGE[3] consistently yields substantially lower negative transfer than SLERP, TIES, and Task Arithmetic across all languages. This indicates that MERGE[3] not only improves average accuracy but also preserves correct knowledge from its component models, thereby maintaining per-example competence during merging.

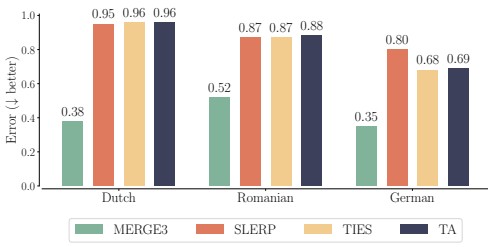

*Figure 15.* Negative Transfer Rate across languages. Lower is better. MERGE[3] shows substantially less negative transfer than standard baselines.

### C.3.3. ADDITIONAL EXPERIMENT: TIME & FLOPS REQUIREMENTS EVOLUTIONARY MERGING

**Hardware Setting.** To compare the efficiency of different model evaluation strategies, we measured the time required to evolve merged LLM models using a single NVIDIA 4090 with 24GB of VRAM, and report the Throughput $R$ in table 9. We also benchmark evaluation and merging times across three GPU models (3090, 4090, V100) to illustrate practical runtimes for MERGE[3] on both modern and older hardware. We report the results in table 10

**Results Discussion.** Over a 12-hour period, we were able to evaluate 8 models on 1000 samples of GSM8K with a single NVIDIA 4090, allowing us to estimate that evaluating 1000 models would take approximately 62 days under similar conditions. In contrast, MERGE[3] enabled the evaluation of a larger number of merged models in significantly less time by using a reduced dataset. These results suggest that researchers and practitioners could leverage consumer-grade GPUs for efficient LLM merging and evaluation, making rapid experimentation of model merging methods more accessible. We report in table 2 the estimated total time of the Evolve runs, which we calculated using the following formula:

$$T(N_{\mathrm{models}}) \;=\; \frac{N_{\mathrm{models}}}{R_{\mathrm{Dataset\ Size}}}$$

Lastly, table 10 shows that MERGE[3] maintains practical runtimes across a range of GPUs. While the 4090 offers the fastest evaluation, older hardware like the V100 still supports feasible experimentation, highlighting the framework's accessibility and the generalizability of results across different consumer GPUs.

**FLOPs Calculation.** We provide a Jupyter Notebook that describes the FLOPs calculations for our experiments in the supplementary material, based on the *calc-flops* li-

*Table 9.* Throughput ($R$) in models per hour for different sample sizes per fitness evaluation on GSM8K. These estimates are based on 12-hour Evolve runs on a single NVIDIA 4090 with 24GB of VRAM.

| Sample size | 1000 | 100 | 50 | 30 | 20 |
|---|---|---|---|---|---|
| **Throughput (Models/Hour)** | 0.67 | 8.33 | 14.17 | 16.67 | 17.08 |

*Table 10.* Evaluation and merge time across different GPU models using Mistral-7B on 10 examples (4-bit, SLERP).

| GPU Model | Eval Time (s) | Merge Time (s) |
|---|---|---|
| NVIDIA 3090 24GB | 65 | 135 |
| NVIDIA 4090 24GB | 45 | 160 |
| NVIDIA V100 32GB | 80 | 220 |

brary[4]. This script has been used to estimate the FLOPs for the experiment in Figure 1.

## D. Mathematical proofs

We outline in table 6 a scheme of the notation used throughout the paper.

### D.1. Proof of Theorem 2

*Proof.* Let $m := F(\theta^*; D)$ and $\hat{m} := F(\hat{\theta}; \bar{D})$. We must show that $|m - \hat{m}| \leq \epsilon$.

1. By $\epsilon$-stability, for *all* $\theta \in \Theta$:

$$\left| F(\theta; D) \;-\; F(\theta; \bar{D}) \right| \;\leq\; \epsilon.$$

In particular, for $\theta = \theta^*$,

$$\left| F(\theta^*; D) \;-\; F(\theta^*; \bar{D}) \right| \;\leq\; \epsilon.$$

Hence

$$F(\theta^*; \bar{D}) \;\geq\; F(\theta^*; D) \;-\; \epsilon$$

and

$$F(\theta^*; \bar{D}) \;\leq\; F(\theta^*; D) \;+\; \epsilon.$$

2. Since $\hat{\theta}$ is the minimizer of $F(\cdot; \bar{D})$, we have

$$F(\hat{\theta}; \bar{D}) \;\leq\; F(\theta^*; \bar{D}).$$

Because $\theta^*$ is the minimizer of $F(\cdot; D)$,

$$F(\hat{\theta}; D) \;\geq\; F(\theta^*; D).$$

---

[4]https://github.com/MrYxJ/calculate-flops.pytorch.

3. To bound $\hat{m} - m$, we can add and subtract $F(\theta^*; \bar{D})$ to have

$$\hat{m} - m = \Big( F(\hat{\theta}; \bar{D}) - F(\theta^*; \bar{D}) \Big)$$
$$+ \Big( F(\theta^*; \bar{D}) - F(\theta^*; D) \Big).$$

The first term is $\leq 0$ (since $\hat{\theta}$ is a minimizer on $\bar{D}$), and the second term is $\leq \epsilon$. Hence

$$\hat{m} - m \leq 0 + \epsilon = \epsilon.$$

4. Analogously, to bound $m - \hat{m}$, we can rewrite

$$m - \hat{m} = \Big( F(\theta^*; D) - F(\hat{\theta}; D) \Big)$$
$$+ \Big( F(\hat{\theta}; D) - F(\hat{\theta}; \bar{D}) \Big).$$

The first term is $\leq 0$ (since $\theta^*$ is a minimizer on $D$), and the second term is $\leq \epsilon$. Thus,

$$m - \hat{m} \leq 0 + \epsilon = \epsilon.$$

5. Combining these inequalities:

$$-\epsilon \leq \hat{m} - m \leq \epsilon \implies |m - \hat{m}| \leq \epsilon.$$

Hence $\big| F(\theta^*; D) - F(\hat{\theta}; \bar{D}) \big| \leq \epsilon$, completing the proof.

$\square$

## D.2. Proof of Theorem 4

*Proof.* By hypothesis, for every $\theta \in \Theta$,

$$\mathbb{E}_{\bar{D}}\Big[ \big| F(\theta; D) - F(\theta; \bar{D}) \big| \Big] \leq \epsilon.$$

Using Jensen's inequality for the absolute value,

$$\big| \mathbb{E}_{\bar{D}}\big[ F(\theta; D) - F(\theta; \bar{D}) \big] \big|$$
$$\leq \mathbb{E}_{\bar{D}}\Big[ \big| F(\theta; D) - F(\theta; \bar{D}) \big| \Big] \leq \epsilon.$$

Hence,

$$-\epsilon \leq \mathbb{E}_{\bar{D}}\big[ F(\theta; \bar{D}) - F(\theta; D) \big] \leq \epsilon$$

for each fixed $\theta$. It thus follows that

$$\mathbb{E}_{\bar{D}}\big[ F(\theta; \bar{D}) \big] \leq F(\theta; D) + \epsilon$$

and

$$\mathbb{E}_{\bar{D}}\big[ F(\theta; \bar{D}) \big] \geq F(\theta; D) - \epsilon.$$

Consequently,

$$\min_{\theta \in \Theta} \mathbb{E}_{\bar{D}}\big[ F(\theta; \bar{D}) \big] \leq \min_{\theta \in \Theta} \big[ F(\theta; D) + \epsilon \big] = m^* + \epsilon,$$

where $m^* := \min_{\theta \in \Theta} F(\theta; D)$. Meanwhile, by a min-versus-expectation (Jensen-type) inequality,

$$\mathbb{E}_{\bar{D}}\big[ \min_{\theta \in \Theta} F(\theta; \bar{D}) \big] \geq \min_{\theta \in \Theta} \mathbb{E}_{\bar{D}}\big[ F(\theta; \bar{D}) \big].$$

Hence,

$$\mathbb{E}_{\bar{D}}\big[ \widehat{m}(\bar{D}) \big] = \mathbb{E}_{\bar{D}}\Big[ \min_{\theta \in \Theta} F(\theta; \bar{D}) \Big]$$
$$\geq \min_{\theta \in \Theta} \mathbb{E}_{\bar{D}}\big[ F(\theta; \bar{D}) \big] \geq m^* - \epsilon.$$

Combining these two bounds results in

$$m^* - \epsilon \leq \mathbb{E}_{\bar{D}}\big[ \widehat{m}(\bar{D}) \big] \leq m^* + \epsilon$$

and, therefore,

$$\Big| m^* - \mathbb{E}_{\bar{D}}\big[ \widehat{m}(\bar{D}) \big] \Big| \leq \epsilon.$$

$\square$

## D.3. Proof of Proposition 5

*Proof.* We must show that

$$\big| \mathbb{E}\big[ \hat{Z}_{jl} \mid Y_{i_0 l}, \dots, Y_{i_k l} \big] - \mathbb{E}\big[ Z_{jl} \mid Y_{i_0 l}, \dots, Y_{i_k l} \big] \big| \to 0$$

in probability as $|\hat{I}| \to \infty$. Under the assumptions of the proposition (including linear inheritance of abilities, $\hat{\lambda} \to \lambda$ in probability, and bounded $\|\alpha_i\|$), we may bound this difference as follows:

$$\big| \mathbb{E}\big[ \hat{Z}_{jl} \mid Y_{i_0 l}, \dots, Y_{i_k l} \big] - \mathbb{E}\big[ Z_{jl} \mid Y_{i_0 l}, \dots, Y_{i_k l} \big] \big|$$
$$\leq \frac{1 - \hat{\lambda}}{|I_j \setminus \hat{I}_j|} \sum_{i \in I_j \setminus \hat{I}_j} \Big| \sigma\big( (\hat{\lambda}_1 \theta_{l_1} + \hat{\lambda}_2 \theta_{l_2})^\top \alpha_i - \beta_i \big)$$
$$- \sigma\big( \theta_{l_m}^\top \alpha_i - \beta_i \big) \Big|.$$

Since $\sigma$ is $1/4$-Lipschitz on $\mathbb{R}$, we have

$$= \leq \frac{1}{|I_j|} \sum_{i \in \hat{I}_j} \Big| \big( (\hat{\lambda}_1 \theta_{l_1} + \hat{\lambda}_2 \theta_{l_2}) - \theta_{l_m} \big)^\top \alpha_i \Big|$$
$$\leq \frac{1}{|I_j|} \sum_{i \in \hat{I}_j} \|\alpha_i\|_2 \| (\hat{\lambda}_1 \theta_{l_1} + \hat{\lambda}_2 \theta_{l_2}) - \theta_{l_m} \|_2.$$

Since $\sup_{i \in I_j} \|\alpha_i\|_2 \leq c$, it follows that

$$\leq c \big\| (\hat{\lambda}_1 - \lambda_1) \theta_{l_1} + (\hat{\lambda}_2 - \lambda_2) \theta_{l_2} \big\|_2 \to 0$$

in probability as $|\hat{I}| \to \infty$. (The last step uses $\hat{\lambda} \to \lambda$ in probability, with $\theta_{l_1}, \theta_{l_2}$ fixed in $\mathbb{R}^d$.) Hence $\hat{Z}_{jl}$ converges in probability to $Z_{jl}$, completing the proof. $\square$

### D.4. Proof of Theorem 2

*Proof.* By Proposition 5, $\hat{Z}^{\text{mp-IRT}}$ becomes arbitrarily close (in probability) to $Z$ as $|\bar{D}| \to \infty$. Under standard regularity conditions, this implies

$$\left| Z(\theta; D) - \hat{Z}^{\text{mp-IRT}}(\theta; \bar{D}) \right| \leq \epsilon$$

in expectation, for all sufficiently large $|\bar{D}|$, hence $\hat{Z}^{\text{mp-IRT}}$ is $\epsilon$-stable in expectation. Applying Theorem 4 completes the argument. $\square$

