# OpenReview forum: "MERGE$^3$: Efficient Evolutionary Merging on Consumer-grade GPUs"
_ICML.cc/2025/Conference — ICML 2025 poster_

### Official Review · Reviewer_iqT4 · 2025-03-11

**Overall Recommendation:** 2

**Summary:**

This paper performs model merging using multi-objective evolutionary search that yields Pareto optimal solutions. The fitness function of the evolutionary algorithm requires one to evaluate a given model’s performance several times. The authors propose using a performance estimator using item-response theory (IRT) that can estimate the model’s true performance by only evaluating it on a subset of the data. NSGA-II evolutionary algorithm is employed by Merge3 where each objective is the performance of the merged model on a particular task. They demonstrate that their method is capable of cross-lingual skill transfer. The report the efficacy of their performance predictor using mean square error between the predicted and the true scores.

**Claims And Evidence:**

Ya the claims are supported by evidence. But like I pointed out in other sections, there is scope for improvement.

**Essential References Not Discussed:**

N/A

**Experimental Designs Or Analyses:**

Please include Ada-merging and Emr-Merging [1] as baselines

[1] EMR-Merging: Tuning-Free High-Performance Model Merging, Huang et al.

**Methods And Evaluation Criteria:**

These make sense
1. This paper demonstrated its efficacy by merging math models with Romanian, German and Dutch langauge respectively and evaluated the merged model's performance on the GSM8k in the corresponding language.
2. They evaluate Merge3's performance against the EvoMerge model using the same settings. Despite using 50 times lesser compute, this method yields a model that is only slightly worse than what EvoMerge resulted in

However, while evaluating performance estimators, it is important to evaluate the spearman correlation between the true ranking and the ranking of the models based on the estimator. The accuracy of the models might be very close to each other. It is important to demonstrate that the model is able to discriminate between them and rank them correctly. This is a common practice in Neural Architecture Search [1], Hyperparameter Optimization etc.

[1] How Powerful are Performance Predictors in Neural Architecture Search? White et al.

**Other Comments Or Suggestions:**

You can spend a paragraph explaining algorithm 1. Also algorithm 1 must be in the main paper and not the appendix.

**Other Strengths And Weaknesses:**

Strengths
1. This work proposes a more efficient way to perform evolutionary search and speeds up evomerge by 50 times.

Weakness
1. Of late, papers such as adamerging, emr-merging etc are merging 8 models each trained on a particular task. Would this method scale to perform well on that task? It would require a lot of more compute as the number of models increase.
2. The IRT based estimators are heavily borrowed from the tinybenchmarks paper with a slight adaptation to estimate only the interpolation coefficients owing to the nature of model merging.
3. What about low resource languages where several models are not available to initialize the population?

**Questions For Authors:**

1. You have demonstrated on cross-lingual tasks. Can your method outperform the baselines on tasks involving the same language?
2. Why are other method performing so poorly on these tasks? Would the surgery [2] model alleviate the representation bias in this case?
3. In algorithm 1, it is not clear how the population is initialized. What is ad-hoc genetic algorithm?

[2] Representation Surgery for Multi-Task Model Merging, Yang et al.

**Relation To Broader Scientific Literature:**

Ya this paper provides an evolutionary algorithm based model merging method that reduces the computational burden by using only a subset of the evaluation data.

**Theoretical Claims:**

Ya I checked their proofs on the performance estimators being $\epsilon$-stable and $\epsilon$-consistent, MP-IRT being asymptotically consistent and also preserving near-optimality.

---

> ### Author Rebuttal · Authors · 2025-04-01
>
> Thank you for your detailed and constructive review. We appreciate your insights and will do our best to address your concerns within the space constraints of the rebuttal.
>
> ### Methods and evaluation criteria
>
> We report Spearman rank correlations (higher is better) using the Figure 3 setup. Due to space limits, we show averages across datasets and sample sizes (10–100); full breakdowns will appear in the paper. Results show that GMP-IRT consistently outperforms GP-IRT in ranking accuracy.
>
> |  | **n=10** | **n=20** | **n=30** | **n=50** | **n=100** |
> | --- | --- | --- | --- | --- | --- |
> | **gmpirt** | **0.54** | **0.68** | **0.73** | **0.83** | **0.84** |
> | **gpirt** | 0.51 | 0.58 | 0.69 | 0.71 | 0.77 |
>
> |  | **arc** | **gsm8k** | **hellaswag** | **truthful** | **winogrande** |
> | --- | --- | --- | --- | --- | --- |
> | **gmpirt** | **0.84** | **0.64** | **0.57** | **0.93** | **0.63** |
> | **gpirt** | 0.77 | 0.59 | 0.48 | 0.92 | 0.51 |
>
> ### Experimental design and analyses
>
> **Ada-Merging and EMR Baselines:** Thank you for highlighting these baselines—we agree they are important. Our current setup relies on MergeKit, which doesn’t yet support them. Implementing and optimizing them (e.g., for quantization) is beyond the rebuttal scope, but we plan to add support and include them in the camera-ready or follow-up.
>
> ### Weaknesses
>
> **W1 — scaling number of models**: Our approach scales well with the number of endpoint models because its computational cost depends on the population size (25 in our experiments) and the number of evolutionary iterations—both independent of how many endpoints are merged. We do note that the search space grows linearly with the number of endpoints, so more sophisticated initialization or mutation strategies might be needed to ensure efficient convergence.
>
> **W2 — novelty of the estimators:** While inspired by tinyBenchmarks, our estimators (mpIRT and gmPIRT) are tailored for merging. By leveraging the known abilities of endpoint models and assuming linearity, we obtain more efficient *and* more accurate ability estimation—crucial for guiding evolutionary search.
>
> **W3 — Low-Resource Settings & Initialization:** MERGE³ uses the same endpoint models as standard merging methods. The initial population is generated by sampling interpolation coefficients—*not* from additional models. This makes our method equally applicable in low-resource settings, and we’ll clarify this distinction in the paper.
>
> ### Comments and questions
>
> **C1:** **Algorithm 1 explanation and position:** We agree Algorithm 1 is central and will move it to the main paper in the camera-ready using the extra page. We'll also add a brief explanation to clarify its key steps.
>
> **Q1 — In-Language Merging:**
>
> While our primary focus was cross-lingual transfer, we also investigated merging models **within a single language**—Italian in this case. We merged a math-in-Italian model ([MetaMath-Mistral-7B + Mistral-Ita-7B](https://huggingface.co/DeepMount00/Mistral-Ita-7b)) with a code model with Italian capabilities ([CodeNinja-1.0-OpenChat-7B](https://huggingface.co/beowolx/CodeNinja-1.0-OpenChat-7B)), then evaluated code generation performance on the [MBPP dataset](https://huggingface.co/datasets/google-research-datasets/mbpp) (zero-shot pass@1) and math accuracy in Italian. As shown below, the merged model not only achieves higher code accuracy but also preserves math performance:
>
> | Model | Code Accuracy | Math Accuracy |
> | --- | --- | --- |
> | Merged Model | **0.218** | **0.596** |
> | Math Model | 0.212 | 0.552 |
> | CodeNinja | 0.200 | 0.192 |
>
> These findings demonstrate that MERGE³ can integrate multiple task-specific abilities within the same language, and we plan to include this experiment in the appendix of the revised paper.
>
> **Q2 — Baselines failure:** We appreciate the reference to [2], which highlights representation misalignment as a major factor in poor performance for standard merging methods. Our baselines typically do not address this issue, so a technique like representation surgery could help reduce bias. While our work focuses on *efficiently* merging models via IRT-based estimators, we view representational alignment as a promising, complementary direction. We will cite [2] in the final version and consider leveraging such approaches to further enhance MERGE³.
>
> **Q3 — Initialization and "Ad-hoc Genetic Algorithm":** We randomly initialize the population by creating interpolations of the same endpoint models used by our baselines—no additional models are needed. By “ad-hoc genetic algorithm,” we mean standard evolutionary operators tailored for merging. Specifically, we use **Simulated Binary Crossover** to recombine parents and **Polynomial Mutation** to introduce small perturbations. We will revise Algorithm 1 and the main text to make these steps clearer.
>
> We thank you again for your valuable feedback. We remain available for any further questions or clarifications.

---

> > ### Comment · Reviewer_iqT4 · 2025-04-08
> >
> > I thank the authors for answering my questions. I would like to keep my score.
> > The paper needs to be clearly written, detailing algorithm 1 including aspects such as the how the population is initialized. As the number of models to be merged increases, as the authors also pointed out, the search space increases. The authors need to demonstrate that their search space and algorithm are capable of yielding a well-performing model. It is also essential to include baselines such as Representation Surgery and Ada-merging and demonstrate that evolutionary search is indeed necessary despite its increased computational expense.

---

> > > ### Author Response · Authors · 2025-04-09
> > >
> > > We thank the reviewer for their response. While we appreciate their engagement, we are disheartened by the outcome, especially given the substantial effort we made to directly address all concerns within the very limited rebuttal window. We believe the raised points were either already addressed or are slated to be fully resolved in the final version of the paper.
> > >
> > > > The paper **needs to be clearly written**, **detailing algorithm 1** including aspects such as the **how the population is initialized**.
> > > >
> > >
> > > We addressed this directly in our rebuttal. **Algorithm 1 has been moved to the main paper with expanded explanations and added clarity.** Due to ICML policy, uploading an updated manuscript during the rebuttal phase was not permitted. Additionally, our supplementary material already includes a code implementation with detailed information about population initialization. We are confident this concern will be fully resolved in the camera-ready version.
> > >
> > > > As the number of models to be merged increases, as the authors also pointed out, **the search space increases**. The authors need to demonstrate that their **search space and algorithm are capable of yielding a well-performing model**.
> > > >
> > >
> > > We respectfully believe this is not a limitation of our approach. In Section 6.3, we merge **four** independently fine-tuned 7B-scale models into a single multilingual model using only a **consumer GPU**—a setting substantially larger and more challenging than those tackled by prior merging methods.  In fact, while EMR-Merging reports *fully* combining 6 language models, each is only GPT-2 Small (124M)—over **70× smaller** per model.
> > >
> > > We also clarify that in evolutionary merging, the **search space grows with the number of merging hyperparameters**, not with model size. For a merge of *n* models, the number of hyperparameters increases linearly, but remains small (e.g., 4 models → 3 coefficients for TIES). The computational bottleneck is not the size of this space, but the **cost of evaluating each candidate model**, which our method drastically reduces via IRT and dataset subsampling. We will expand on this explanation in the final version.
> > >
> > > > It is also essential to include baselines such as **Representation Surgery and AdaMerging** and **demonstrate that evolutionary search is indeed necessary** despite its increased computational expense.
> > > >
> > >
> > > We respectfully **disagree that these baselines are essential in the context of generative large-scale LLM merging**. Both Representation Surgery and AdaMerging were designed for computer vision tasks, and their use in NLP is limited to small models such as BERT or GPT-2 Small. These methods are not implemented in MergeKit (the de facto LLM merging library), nor are they used in the **Hugging Face Open LLM Leaderboard**, which includes nearly 5,000 models—many of them merged using the baselines that we included. Moreover, adapting AdaMerging or Representation Surgery to our setting would **require substantial engineering** effort to scale them from 100M–200M parameter models to 7B+ and **is beyond the scope of this work, likely warranting a dedicated paper in its own right.** This was not feasible within the tight rebuttal window, especially given the other new experiments we included (e.g., Spearman correlation, in-language merging). We fully intend to explore these methods in future work or the camera-ready version.
> > >
> > > If these concerns played such a central role in the reviewer’s final decision, we believe it would have been appropriate to explicitly state their impact on the final score earlier in the discussion—rather than only a few hours before the discussion period closes and **4 days** **after the acknowledgment deadline had passed.** Given the limited timeframe, we feel we were unfairly penalized without a fair opportunity to respond. As noted previously, we plan to support and include these baselines in the camera-ready version or a follow-up submission, while all the other concerns have been addressed.
> > >
> > > In summary, we believe we have responded comprehensively and constructively to all points raised. We respectfully ask the reviewer to reconsider their score, as the remaining concerns do not appear to constitute grounds for rejection given the contributions and evidence provided. We remain committed to further improving the paper and incorporating all helpful feedback in the final version.

---

### Official Review · Reviewer_XxKs · 2025-03-11

**Overall Recommendation:** 3

**Summary:**

This paper proposes an efficient evolutionary model merging framework to achieve multilingual model merging and cross-language knowledge transfer, and conducts a large number of experiments and theoretical analysis to verify the effectiveness of the method.

**Claims And Evidence:**

YES

**Essential References Not Discussed:**

Clarify the more specific challenges/differences between [1] and this paper's method.

[1] tinybenchmarks: evaluating llms with fewer examples. ICML, 2024.

**Experimental Designs Or Analyses:**

YES

**Methods And Evaluation Criteria:**

YES

**Other Comments Or Suggestions:**

See weaknesses

**Other Strengths And Weaknesses:**

Strengths:
- This paper proposes an efficient evolutionary model merging framework for multilingual model merging and cross-lingual knowledge transfer.
- This paper provides theoretical guarantees to verify the effectiveness of the method.
- This paper is clearly written and provides source code implementation.

Weaknesses:
- This paper seems to be just an application of [1] to model merging. It is not clear what challenges/difficulties there are in directly applying [1] to model merging, as this seems to be a straightforward application. The authors mention [1] in all subsections of this paper’s methods section, which seems to be a direct extension of [1] in the model merging setting. The authors need to clarify the connection more deeply.
- This paper relies on a large validation set for data selection, however, in the standard model merging setting, such data does not seem to be available.
- If validation set data is available, how does the performance compare to the approach of this paper if the available validation data is directly used as input to the test phase of the merged model through in-context learning?

[1] tinybenchmarks: evaluating llms with fewer examples. ICML, 2024.

**Questions For Authors:**

See weaknesses

**Relation To Broader Scientific Literature:**

This paper significantly improves the efficiency of previous evolutionary model merging[1].

[1] Evolutionary optimization of model merging recipes. Nature Machine Intelligence, 2025.

**Theoretical Claims:**

YES

---

> ### Author Rebuttal · Authors · 2025-04-01
>
> Thank you for your thorough evaluation of our work and for highlighting areas in need of clarification. We appreciate the opportunity to provide more detail on our key contributions relative to [1], our assumptions regarding the availability of a validation set, and our comparisons with in-context learning (ICL). Below, we address each of your points in turn.
>
> ### Weaknesses
>
> **W1** — **difference with tinyBenchmarks:** We thank the reviewer for pointing this out and agree that it is essential to clarify the difference from [1] (“tinyBenchmarks”).
>
> While [1] provides two estimators (pIRT and gpIRT) for efficient *evaluation* of large language models, our work proposes two *new* estimators—mpIRT and gmPIRT—***specifically tailored for model merging***. We demonstrate that plugging these new estimators into the *evolutionary* merging pipeline yields a *fifty-fold* reduction in computational costs, with negligible loss in accuracy.
>
> Concretely, mpIRT and gmPIRT incorporate the *additional prior* we have in the merging setting, namely ***access to all endpoint models’ abilities***. This lets us approximate the merged model’s ability by a *linear combination* of the endpoints’, rather than fitting a full IRT vector from scratch. Doing so drastically reduces the data and compute needed per iteration of evolutionary search, which is the main bottleneck. By contrast, directly applying [1] to merging *without* leveraging this extra prior would require re-fitting a merged model’s IRT parameters each time, significantly slowing down evolution. Thus, although we take inspiration from [1]’s general approach to *efficient performance estimation*, our paper addresses new challenges in model merging, such as evolving interpolation coefficients across multiple tasks/languages and making performance estimators *merging-aware*.
>
> **W2 — Need for a validation set:** We follow the evolutionary merging setting introduced in [2], which assumes a validation (or evaluation) set is available to measure candidate models’ fitness. This convention is also common in other merging works, such as those that use a held-out set to tune the scaling factor [3]. That said, we agree that exploring *fully unsupervised* proxies—e.g., perplexity or entropy on unlabeled data—would be an exciting direction, as it would relax the requirement for supervised validation data in merging. We plan to investigate such approaches in future work.
>
> [1] Polo, Felipe Maia, et al. "tinyBenchmarks: evaluating LLMs with fewer examples." *International Conference on Machine Learning*. PMLR, 2024.
>
> [2] Akiba, Takuya, et al. "Evolutionary optimization of model merging recipes." *Nature Machine Intelligence* (2025): 1-10.
>
> [3] Ilharco, Gabriel, et al. "Editing models with task arithmetic." *The Eleventh International Conference on Learning Representations*.
>
> **W3 — Comparison with In-Context Learning (ICL)** We appreciate the reviewer’s suggestion. To address it, we ran the proposed few-shot in-context learning approach on our multilingual experiments, providing 20 samples as context at inference time for two baselines (TIES-DARE and Task Arithmetic). As shown in the table below, Merge³ significantly outperforms these few-shot baselines on each language:
>
> | Method | DE | IT | NL | EN |
> | --- | --- | --- | --- | --- |
> | TIES-DARE Few-shot (20) | 0.227 | 0.226 | 0.227 | 0.226 |
> | Task Arithmetic Few-shot (20) | 0.427 | 0.406 | 0.491 | 0.566 |
> | Merge³ (gmpIRT-20) [Ours] | **0.720** | **0.690** | **0.690** | **0.790** |
>
> Furthermore, using ICL makes the context significantly longer and increases memory requirements, whereas our merged model has *no additional overhead at inference*. Once we merge the models offline, the resulting single network can be deployed with the same resource footprint as a standard model of that size. Thus, while few-shot ICL can help in certain scenarios, **model merging** provides a more permanent and resource-efficient solution.
>
> We are grateful for the time and attention you invested in reviewing our paper. Your feedback has been very helpful, and we believe that the clarifications provided address your concerns. We look forward to further refining our work in response to your comments.

---

### Official Review · Reviewer_ysfV · 2025-03-15

**Overall Recommendation:** 4

**Summary:**

This paper introduces MERGE3, a framework for efficient evolutionary model merging on consumer-grade GPUs. The method addresses computational bottlenecks in evolutionary merging by: (1) extracting a reduced dataset for evaluation, (2) estimating model abilities using Item Response Theory (IRT), and (3) evolving optimal merges via IRT-based performance estimators. The authors assert MERGE3 reduces fitness computation costs by 50× while preserving performance.

**Claims And Evidence:**

The paper makes several claims that are generally well-supported:
1. The 50× reduction in computational cost is demonstrated through calculations and empirical measurements.
2. Performance preservation is shown through comparisons with models evolved on full datasets.
3. Cross-lingual knowledge transfer effectiveness is demonstrated through experiments on multiple language pairs.

The evidence is convincing, particularly the experimental results showing comparable performance to models requiring much greater computational resources.

**Essential References Not Discussed:**

The literature review is comprehensive, covering key works in model merging, evolutionary algorithms, and IRT.

**Experimental Designs Or Analyses:**

The experiments are comprehensive but have some limitations:
1. Good coverage of cross-lingual transfer and multilingual model evolution
2. Appropriate baselines and metrics
3. Limited analysis of hyperparameter sensitivity
Ablation studies could be more extensive to isolate the contribution of each component
Hardware benchmarks are limited to a single GPU model (NVIDIA 4090)

**Methods And Evaluation Criteria:**

The methodology is coherent and properly described. The three-stage approach (Extract, Estimate, Evolve) is logically structured. Evaluation metrics and baselines are appropriate. The authors rigorously evaluate against standard merging techniques (TIES-DARE, SLERP, Task Arithmetic) and compare against state-of-the-art models like EvoLLM-JP-7B.

**Other Comments Or Suggestions:**

1. Consider expanding the analysis to more GPU configurations

**Other Strengths And Weaknesses:**

Strengths:
1. Addresses a practical limitation in state-of-the-art model merging
2. Novel theoretical guarantees for performance estimation
3. Open-source library implementation

Weaknesses:
1. Limited discussion of potential negative transfer in cross-lingual merging
2. Could explore more dataset reduction strategies beyond random sampling

**Questions For Authors:**

How does your approach extend to scenarios with more than two endpoint models?

**Relation To Broader Scientific Literature:**

The paper properly positions itself within the model merging literature. Connections to IRT are well-established, and the authors appropriately attribute previous work

**Theoretical Claims:**

The theoretical foundation is solid. The authors provide formal guarantees for their performance estimators.

---

> ### Author Rebuttal · Authors · 2025-04-01
>
> We thank you for your thoughtful and detailed review. We are glad you find our framework coherent, our theoretical underpinnings solid, and our empirical results convincing. Below, we address your specific points and questions.
>
> **Limited analysis of hyperparameters:** The main hyperparameter in our pipeline is the interpolation coefficient **c** (Equation 5), for which we followed the default from TinyBenchmarks [1] without tuning.
>
> Following the reviewer’s suggestion, we ran a hyperparameter sweep over **c**. The results (included in the revised paper) show that while our original methods—**MP-IRT** and **GMP-IRT**—already outperformed competitors, the tuned **GMP-IRT*** further improves performance
>
> | Dataset | GMP-IRT* | GMP-IRT | GP-IRT* | GP-IRT | MP-IRT |
> | --- | --- | --- | --- | --- | --- |
> | ARC | **0.035**  | 0.040  | 0.046  | 0.049 | 0.048 |
> | WINOGRANDE | **0.018**  | 0.031  | 0.032  | 0.037 | 0.036 |
> | GSM8k | **0.057** | 0.057  | 0.074  | 0.064 | 0.062 |
> | HELLASWAG | **0.046** | 0.056  | 0.077  | 0.071  | 0.047 |
> | TRUTHFULQA | **0.040**  | 0.045  | 0.062  | 0.055  | 0.044 |
>
> For the **evolutionary run**, we used 175 individuals, split into 25 subpopulations over 7 iterations—chosen to ensure all experiments completed in under 24 hours. While not extensively tuned, this setup balanced runtime and performance well.
>
> **Ablation studies:**  Because evolutionary search is inherently noisy and computationally expensive to run many times end-to-end, the most feasible way to measure each module’s impact is to isolate it and evaluate it independently. In Figure 3, we focus on performance estimators, while Figure 4 examines ability estimators. Additional ablation results are included in Appendix C.2 and C.3. We agree these analyses are crucial, and we will add a concise summary of them in the main text to more clearly highlight each component’s contribution.
>
> **Benchmarks limited to one GPU.** We will include multi-GPU benchmarks in the revised paper to better illustrate MERGE³’s accessibility. Below is an example using Mistral-7B on GSM8K-RO (10 examples, 4-bit models, SLERP merging):
>
> | **GPU** | **Eval Time** | **Merge Time** |
> | --- | --- | --- |
> | 3090 24GB | 65s | 135s |
> | 4090 24GB | 45s | 160s |
> | V100 32GB | 80s | 220s |
>
> These times show MERGE³ is practical even on older GPUs. We will add more results in the final version, including how runtimes scale with GPU class and batch size.
>
> ### Weaknesses
>
> **W1 — negative transfer:** In our updated experiments, we study **negative transfer** in cross-lingual merging using the DE, NL, RO GSM8K dataset and compare the **Negative Transfer Rate (NTR)** of MERGE3 against SLERP, TIES, TA. Negative transfer is observed when the merged model fails to answer a question correctly despite at least one base model having answered it correctly. Namely, `NTR = (# of negatively transferred questions) / (# of questions at least one base model got right)`
>
> | Language | MERGE3 (↓ better) | SLERP | TIES | TA |
> | --- | --- | --- | --- | --- |
> | Dutch | **0.38** | 0.95 | 0.96 | 0.96 |
> | Romanian | **0.52** | 0.87 | 0.87 | 0.88 |
> | German | **0.35** | 0.80 | 0.68 | 0.69 |
>
> We see that MERGE3 substantially reduces negative transfer compared to standard interpolation methods. We will include full details and derivations in the appendix for transparency.
>
> **W2 — additional data reduction strategies**: We agree that exploring non-random sampling strategies is an intriguing direction, and we experimented with two additional methods—an IRT-based clustering approach (as in [1]) and a “Representation Clustering” technique that uses concatenated embeddings from our endpoint models and applies PCA plus k-means. In both cases, we observed no clear performance benefits relative to simpler random sampling, especially after considering the added complexity and compute overhead. Thus, we ultimately opted for random sampling, but we will include these findings (presently in Appendix C.1) more prominently in the final version, highlighting why more sophisticated methods did not yield sufficient gains to justify their complexity.
>
> ### Questions
>
> **Q1 — extending to more than two endpoints**: Our evolutionary framework naturally extends to merging more than two endpoint models. The only change is an increase in the dimensionality of the search space, as we optimize more interpolation coefficients. The rest of the pipeline remains unchanged. We maintain a fixed population size (e.g., 25) and apply standard evolutionary operations. We demonstrated this in practice by merging three models for Jap math and four models across IT, EN, NL, and DE in the multilingual setting—both without any changes to the architecture or training procedure.
>
> Thank you again for your thoughtful feedback. We're happy to provide further clarification if needed.

---

### Official Review · Reviewer_sVdP · 2025-03-17

**Overall Recommendation:** 3

**Summary:**

The authors present a framework for efficient evolutionary merging of language models for creating models with strong multi-task and/or cross-lingual task performance from a library of existing fine-tuned models without additional fine-tuning. In MERGE$^3$, the critical efficiency benefit comes from Extracting (in this case, randomly sampling) a much smaller sample of examples for different tasks from full datasets. Latent ability vectors are then iteratively Estimated for each model on different tasks via Item Resposne Theory, and these IRT-based estimators are used to inform optimal merges for each iteration. The authors conclude that MERGE$^3$ results in models that are competitive with an evolutionary merging algorithm that relies on full evaluation at each step.

**Claims And Evidence:**

The paper's claims are generally supported when they are are concrete and specific. To me, however, it feels like a stretch to state that the algorithm "reduc\[es\] fitness computation costs 50x **while preserving performance**". In my opinion, it would be better to state objectively the % of accuracy performance preserved at the fraction of compute used. Additionally, the concrete example and general description (at least VRAM capacity) of the consumer GPU used should be included earlier than in the Appendix.

**Essential References Not Discussed:**

n/a

**Experimental Designs Or Analyses:**

seems sound and valid.

**Methods And Evaluation Criteria:**

Overall yes, though some aspects of presentation could be improved for fairness and clarity. I would have liked to see wall-clock training time results presented in the main body -- presumably there might be some amount of overhead from loading in models and data such that naive estimation from FLOP counts may be insufficient

**Other Comments Or Suggestions:**

1. Consider updating the title to have "Evolutionary **Model** Merging", and specifically mentioning language models in the abstract
2. In the intro, somewhere between "In this paper" and "Our approach", I would have liked to see it explicitly stated that fitness computation is a key bottleneck in the standard evolutionary merging approach that is being compared to. Though it is clear looking back, it could help with clarity on a first read
3. "Consumer GPU" can mean many different things -- I think it would improve the specificity of the authors' claims to state explicitly the kind of GPU that was used and what kind of VRAM it has, earlier in the main body. Currently it is only implied that the specific device that benchmarking was performed on was a 4090, and GPU's capacity is not stated until the appendix.
4. I would have liked to see something like Table 7 in the main body
5. I would strongly prefer avoidance of using the same $i$ to index variables repeatedly. It sometimes refers to examples in $\mathcal{D}$ (e.g. under Eq.1), but it sometimes seems to refer to models in the pool (e.g. in Eq. 3). Is Assumption 1 meant to imply that the dimension of the latent ability vectors must equal $|\mathcal{D}|$?
6. Please define "endpoint model" before using it, and spell out MP-IRT and GMP-IRT the first time it appears
7. In general, aligning axes/scales across figures in the same group would help with clarity. For Figure 4 in particular, going from 0 to 1.0 in the same increments would be helpful

**Other Strengths And Weaknesses:**

Strengths:
1. Reported effiicency gains are significant and do indeed enable an approximation of a method that is prohibitively expensive if performed naively
2. Interesting use of item response theory in the context of model merging
3. Proposed framework is supported by both empirical results and theoretical grounding

Weaknesses:
1. See Questions 2-4. I have some uncertainty about the true wall-clock compute savings that result from using MERGE$^3$. I am thinking of e.g. potential overhead from repeated loading of different models into VRAM. Though these numbers are dependent on the specific GPU, model, and amount of data, I would feel better about accepting the "50x" figure with some kind of breakdown of specific amount of time needed for each framework step (and for each end-to-end iteration of Evolve)
2. Some clarity issues throughout, especially in the Section 5 theoretical analysis. See Comments 5-6 below
3. Missing baseline , See Q6

**Questions For Authors:**

1. Did the authors try any merging in data flow space as Akiba et al did?
2. What exactly is meant by "Estimated total time" in Table 7? Does this only account for Evolve runs (not Estimating?)
3. Relatedly, is my understanding of Estimate vs Evolve correct? I understand Estimate as an initial, more complete computation of estimated ability in the initial candidate pool of models. Estimate is only performed once. Evolve, then, also covers iterative updating of performance estimator parameters via repeated inference on the *reduced* set of examples from the Extract step -- is this correct?
4. Can the authors provide a discussion of scaling of compute requirements as datasets, set of tasks, and/or models grow, especially wrt Estimate vs Evolve? e.g. is there some order of magnitude of dataset size at which the compute requirements of Estimate could be expected to dominate Evolve?
5. What is the expected dimensionality of the latent ability vectors?
6. In Table 1, authors provide a baseline of translated ARC performance on models only fine-tuned for the language. Are there baselines for translated ARC performance on models fine-tuned only for ARC? Additionally, I understand that this is not always practical, but in this case it seems possible to provide a baseline of translated ARC performance on models fine-tuned on the translated ARC datasets, along with information about compute requirements for the same. This would help us understand whether MERGE$^3$ is expected to pareto-dominate even a direct fine-tuning approach when sufficient training data is available

**Relation To Broader Scientific Literature:**

Main related work seems to be the Akiba et al Evolutionary Model Merging paper, where the key contribution over that paper is to use only a much smaller subset of dataset examples to estimate latent model ability.

**Theoretical Claims:**

Seems correct, but I have some issues with clarity (see below)

---

> ### Author Rebuttal · Authors · 2025-04-01
>
> Thank you for the detailed and thoughtful review. Your feedback helped us identify key areas for clarification. Below, we respond to each point.
>
> **Claims And Evidence**
>
> - In the revision, we’ll replace “while preserving performance” with: 50× compute reduction with ~86% accuracy retained (e.g., Japanese GSM8K). The 50× figure reflects general FLOP savings; the 86% retention is task-specific and may vary. We’ll clarify this distinction to avoid overstatement.
> - We will clarify the GPU used for the experiments in the main manuscript, and specify minimum VRAM requirements. We used an RTX 4090, with Batch Size 8, Quantization 4bit, Model size 7B.
>
> **Methods And Evaluation Criteria**
>
> - Please refer to W1.
>
> ### Weaknesses
>
> - **W1 — true wall clock time**: We agree that FLOPs don’t fully capture runtime efficiency. We’ll include wall-clock breakdowns of Estimate and Evolve steps, as well as end-to-end timing per MERGE³ iteration. To clarify: both MERGE³ and EvoMerge load the same models, so load time is similar; the gain comes from evaluating fewer examples, which significantly reduces total runtime. We’ll support this with empirical data.
> - **W2 — clarity issues:**  We now use *i* for data examples and *j* for models to remove ambiguity. “Endpoint model” is defined at first use, and MP-IRT/GMP-IRT are spelled out with brief explanations. Assumption 1 does not imply a latent space of dimension |D|, but that a merged model’s ability is a linear combination of endpoint abilities. We’ll clarify its role further in the revised paper.
> - **W3 — finetuning baseline**: Unfortunately, full fine-tuning of a 7B model on ARC isn’t feasible within the rebuttal. We also note that this experiment has substantially different requirements from our budget-friendly MERGE3 framework, which is specifically designed for consumer-grade hardware and operates using only a small number of datapoints (20). That said, we agree this comparison would be valuable and plan to include it in the final version.
>
> ### Comments
>
> **C1 — changing title:** we’ve updated the abstract to explicitly mention that our method targets **language models**, which we agree improves clarity. Regarding the title, we’ll aim to revise it to include “evolutionary *model* merging” for the camera-ready version, subject to the venue’s guidelines on title changes.
>
> **C2 — emphasizing fitness as a key bottleneck**: We revised the introduction to highlight that **fitness evaluation is the main bottleneck** in evolutionary merging.
>
> **C3 — GPU specification**: Please see the “claims and evidence” section.
>
> **C4 — wall-time**: Please see the answer to W1.
>
> **C5 and C6 —** **Clarifying notation:** Thanks for pointing out this lack of clarity. Please refer to the answer to W2.
>
> **C7 — Aligning axes in figures:** we will align the axes and scales across grouped figures so that all plots use consistent increments in the revised version of the paper.
>
> ### Questions
>
> **Q1 — Merging in Data Flow Space (DFS)**: We tried DFS merging as in Akiba et al., but found no consistent gains. Their Table 1 shows DFS often underperforms parameter space (PS) merging, despite adding 3B parameters—nearly half the base model. Given our focus on efficiency for consumer GPUs, this overhead was impractical, so we prioritized size-preserving strategies.
>
> **Q2 — Estimated total time**: It is a measure based on up to 12 hours runs for each method on a single NVIDIA 4090. This is an end to end measure of the entire pipeline: including model loading, estimation of abilities, and evolution of models.
>
> **Q3 — understanding of Estimate vs Evolve**: Yes, that’s correct. **Estimate** is run once to compute endpoint abilities using the full dataset. **Evolve** then runs iteratively, using only the reduced dataset and our estimators to evaluate new merged models efficiently.
>
> **Q4 — scaling of compute requirements**: We agree this is an important consideration and will include a discussion in the paper. In MERGE³, **Estimate** is run once per endpoint model on the full dataset, while **Evolve** runs repeatedly on a reduced subset. In practice, **Evolve dominates compute** because:
>
> 1. **Correctness labels are often public** (e.g., Open LLM leaderboard), making Estimate nearly free.
> 2. **Estimate is one-time**, whereas Evolve runs over many generations and a full population.
>
> When correctness must be computed manually, a rough rule of thumb is: **if** M × N > P × K (where M = endpoint models, N = full dataset size, P = population, K = reduced subset), then Estimate might dominate. Otherwise, Evolve is the primary cost.
>
> **Q5 — dimensionality of the latent ability vectors**: The dimensionality is a hyperparameter; we set it to **16**, following TinyBenchmarks [1].
>
> **Q6 — see W1**
>
> We thank you again for your valuable feedback. We remain available for any further questions or clarifications.

---

> > ### Comment · Reviewer_sVdP · 2025-04-05
> >
> > Thank you, I appreciate the detailed response. I would be happy to see this paper be accepted. Looking forward to see future iterations

---

> > > ### Author Response · Authors · 2025-04-09
> > >
> > > Thank you again for your thoughtful and constructive feedback. We’ve done our best to address your concerns in the rebuttal and are planning additional improvements in the final version. If you feel your concerns have been resolved, we’d be grateful if you would consider updating your score.

---

### Decision · Program_Chairs · 2025-05-01

**Decision:**

Accept (poster)

**Comment:**

This paper presents a method for reducing the evaluation time in evolutionary model merging. The proposed method, MERGE^3, extracts a reduced dataset for evaluation and estimates the model abilities using item response theory (IRT). The idea of reducing technique is inspired by existing work of tinybenchmarks. The effectiveness of the proposed method is demonstrated through several model merging experiments.

The effectiveness of the proposed method is supported by experiments and theoretical analysis, which is an advantage of the paper. Also, the open-source library will be useful for the community.

In the original manuscript, there are several unclear and overclaiming descriptions, which have been pointed out by reviewers and adequately addressed in the rebuttal phase. These descriptions should be revised accordingly in the revised manuscript.

In addition, I encourage you to include the result of the wall-clock training time and the computation breakdown, including the overhead of model loading, in the main text.